# A Closed-Form Solution for Fast and Reliable Adaptive Testing

**Yan Zhuang**[1], **Chenye Ke**[2], **Zirui Liu**[1], **Qi Liu**[1,3],[*] **Yuting Ning**[4], **Zhenya Huang**[1,3],
**Weizhe Huang**[1], **Qingyang Mao**[1], **Shijin Wang**[5]

1: State Key Laboratory of Cognitive Intelligence, University of Science and Technology of China
2: Anhui University 3: Institute of Artificial Intelligence, Hefei Comprehensive National Science Center
4: Ohio State University 5: iFLYTEK Co., Ltd
`{zykb,liuzirui,hwz871982879,maoqy0503}@mail.ustc.edu.cn,`
`kecy@stu.ahu.edu.cn, ning.151@osu.edu,`
`{qiliuql,huangzhy}@ustc.edu.cn, sjwang3@iflytek.com`

## Abstract

Human ability estimation is essential for educational assessment, career advancement, and professional certification. Adaptive Testing systems can improve estimation efficiency by selecting fewer, targeted questions, and are widely used in exams, e.g., GRE, GMAT, and Duolingo English Test. However, selecting an optimal subset of questions remains a challenging nested optimization problem. Existing methods rely on costly approximations or data-intensive training, making them unsuitable for today's large-scale and complex testing environments. Thus, we propose a Closed-Form solution for question subset selection in Adaptive Testing. It directly minimizes ability estimation error by reducing ability parameter's gradient bias while maintaining Hessian stability, which enables a simple greedy algorithm for question selection. Moreover, it can quantify the impact of human behavioral perturbations on ability estimation. Extensive experiments on large-scale educational datasets demonstrate that it reduces the number of required questions by 10% compared to SOTA methods, while maintaining the same estimation accuracy.

## 1 Introduction

Accurate assessment of human abilities plays a crucial role in education, career advancement, and professional certification, directly influencing future opportunities. As a result, the demand for effective and efficient assessment methodologies has grown significantly [1, 2, 3]. Traditional paper-and-pencil tests require examinees to answer a large number of questions, leading to cognitive load and inefficiency. In contrast, Adaptive Testing has emerged as a highly efficient ability estimation approach and has been widely adopted in education systems, and has been successfully integrated into various standardized testing systems [4, 5].

The effectiveness of adaptive testing lies in a key insight: not all questions are equally valuable for estimating ability. To achieve efficiency while maintaining accuracy, an adaptive testing system relies on two key components: 1) Question selection algorithm – Identifying and selecting the most informative subset of questions from the full question pool; 2) Item Response Theory (IRT) – A psychometric framework [6] that models the relationship between an examinee's latent ability $\theta$ and their observed responses (correct/incorrect). IRT serves as the "user model" for estimating ability based on response data to the selected questions.

---

[*]Corresponding Author.

39th Conference on Neural Information Processing Systems (NeurIPS 2025).

From a machine learning perspective, the overall adaptive testing process can be formulated as *a subset selection problem that seeks to minimize the error of ability estimation* [7, 8]: Given a large question pool $V$, selecting a question subset $S \subseteq V$ for an examinee to answer such that the ability estimate $\theta_S$ (inferred from responses to $S$) is as close as possible to the true (or optimal) ability $\theta^*$:

$$\min_{S \subseteq V} \|\theta_S - \theta^*\|, \quad \text{s.t.} \quad \theta_S = \arg\min_{\theta \in \Theta} \sum_{i \in S} \ell_i(\theta), \tag{1}$$

where $\ell_i(\theta)$ denotes the cross-entropy loss associated with the response to question $i$, and $\theta$ represents the ability parameter modeled by IRT. Obviously, adaptive testing is a complex *nested optimization* w.r.t. the subset variable $S$, requiring iterative updates: the outer loop selects the optimal subset $S$ (often represented as a sparse selection vector [8]), while the inner loop estimates the ability parameter via supervised learning.

Given its complexity, recent works often rely on data-driven meta-learning [9], or reinforcement learning [10, 2] to derive the question selection policy. However, these approaches introduce significant computational overhead and may amplify biases present in the data [9]. Even latest heuristic algorithms [7, 11] still require approximating and matching gradients across the entire ability parameter space $\Theta$, leading to prohibitively high complexity. These limitations are especially critical in real-world online assessments, e.g., the Duolingo English Test, GRE Online, and remote certifications, which involve massive item pools, diverse examinees, and complex user behavior. These settings demand adaptive testing systems that are interpretable, robust, and efficient enough for real-time operations [12, 13].

To address these, this paper proposes a fundamental shift in the optimization paradigm of adaptive testing. For the first time, we derive a closed-form solution for the unknown subset variable $S$, referred to as CFAT (Closed-Form expression for Adaptive Testing). It allows us to directly solve for the optimal subset without iterative sampling or complex nested optimization. Specifically, we successfully quantify the ability estimation error and demonstrate that it can be interpreted as minimizing the gradient bias while maintaining a stable Hessian structure. Furthermore, we prove that the objective function exhibits approximate submodularity, enabling the use of a simple greedy algorithm to efficiently select the subset.

Beyond improving question selection, such closed-form formulation allows us to quantify the impact of human behavioral perturbations (e.g., guessing and slipping) on ability estimation. CFAT ultimately enables a bias correction mechanism for more reliable assessments. By fundamentally shifting the optimization paradigm of adaptive testing, CFAT uses statistical learning principles for efficient, direct computation. Experiments on three educational datasets demonstrate that our method reduces the number of required test questions by 10% compared to the best baseline, under the same estimation accuracy. Moreover, CFAT achieves at least a 12× improvement in selection efficiency (computation time) over latest methods. It can also exhibit higher robustness in high-noise scenarios, accurately recovering ability estimates and improving prediction reliability.

## 2 Background and Related Works

Adaptive testing has been widely adopted in human ability assessment especially in education, and has gradually been incorporated into high-stakes examinations. To achieve both accuracy and efficiency, adaptive testing typically consists of two key components: IRT and question selection algorithms:

*(1) Item Response Theory (IRT).* IRT serves as a user model that captures the relationship between an examinee's ability and their responses [4]. Widely used in various large-scale assessments such as OECD/PISA, a common example is the two-parameter logistic (2PL) model, which defines the probability of a correct response to question $i$ as: $p(\text{correct}) = \sigma(\alpha_i(\theta - \beta_i))$, where $\alpha_i$ and $\beta_i$ represent the discrimination and difficulty parameters, respectively. These question parameters are pre-calibrated [14], while the examinee's ability $\theta$ is estimated during testing. IRT models are interpretable: higher ability implies higher probability of success on items of fixed difficulty. Extensions include multidimensional IRT [15] and neural cognitive diagnosis models [16, 17, 18], which capture more complex interactions. All these methods rely on maximum likelihood estimation (minimizing cross-entropy loss) to estimate ability parameters from observed response data.

*(2) Selection Algorithm.* This is the core of achieving efficient assessment, as it determines a valuable subset for estimating examinee ability in IRT. Traditional algorithms rely on statistical heuristics

based on information measures, such as Fisher information [14], KL information [19] and various improved information metrics [20, 21, 22], to guide selection. Alternatively, active learning methods select informative questions based on question diversity and uncertainty [23]. Recently, to directly solve the nested optimizations, researchers have increasingly adopted data-driven approaches, e.g., reinforcement learning and meta-learning, to optimize subset selection [10, 9, 2, 8]. These methods iteratively train a policy (often represented as a neural network) from large-scale response data.

In this work, we aim to bypass the nested optimization by deriving a closed-form expression for the estimation error w.r.t. the selected subset. It allows us to determine the optimal subset directly. Compared to data-driven/neural network-based approaches, this statistical method eliminates the need for extensive training. Compared to the latest gradient-based heuristic algorithms [7, 11], it incorporates second-order gradient (Hessian matrix) information, meanwhile, mitigating the impact of guessing and mistakes on ability estimation. Furthermore, CFAT is up to 12× more efficient than these SOTA heuristic methods, making it highly practical for real-time human assessments.

## 3 Method

Adaptive testing estimates ability efficiently by selecting a small, informative subset $S \subseteq V$ from a larger question pool $V$. It can reduce test length while maintaining accuracy.

**Problem Statement.** Formally, an examinee responds to the selected subset $S$, producing $\{(q_1, y_1), ..., (q_{|S|}, y_{|S|})\}$, where $S = \{q_i\}_{i=1}^{|S|} \subseteq V$ is the question set selected by the adaptive selection algorithm, and $y_i \in \{0, 1\}$ denotes the response label, with 1 representing a correct response and 0 otherwise. The examinee's ability is then estimated by minimizing cross-entropy loss $\ell$ over $S$.

$$\theta_S = \arg\min_\theta \sum_{i \in S} \ell_i(\theta) = \arg\min_\theta \sum_{i \in S} -\log p_\theta(q_i, y_i), \tag{2}$$

where $p_\theta(q_i, y_i)$ represents the probability of observing response $(q_i, y_i)$ from an examinee with ability $\theta$. The precise form of $p_\theta$ depends on the IRT model. Assuming an examinee's true latent ability is denoted as $\theta^*$, one can theoretically approximate it by minimizing the expected cross-entropy loss over the entire question pool: $\theta^* = \arg\min_\theta \sum_{i \in V} \ell_i(\theta)$ [7]. The objective of adaptive testing is to ensure that the estimated ability $\theta_S$ from the subset is as close as possible to $\theta^*$ (Figure 1):

**Definition 1** (Definition of Adaptive Testing). *Given a fixed test length $T$, the task is to select an optimal subset $S \subseteq V$ such that the ability estimate $\theta_S$ approximates the estimate $\theta^*$. The adaptive testing task can be formulated to a nested optimization problem as follows::*

$$\min_{|S|=T} \|\theta_S - \theta^*\|, \quad s.t. \quad \theta_S = \arg\min_\theta \sum_{i \in S} \ell_i(\theta). \tag{3}$$

In the *outer loop*, the subset $S$ can be generated using a selection policy $\pi$ [10, 9, 2], or it can be treated as a trainable indicator or sparse selection vector that determines question selection [8]. In the *inner loop*, a base optimization algorithm estimates $\theta_S$ using the responses on the selected $S$, following standard supervised learning principles.

While reinforcement/meta-learning methods have shown promise in adaptive testing [1], they are often computationally intensive due to multi-step gradient descent and repeated backpropagation. This raises: *Can we directly formulate $\|\theta_S - \theta^*\|$ and optimize $S$ without resorting to iterative meta-optimization?* If the effect of question selection on ability estimation can be explicitly modeled, more efficient selection strategies may be possible.

### 3.1 Avoid the Nested Optimization Trap

The key challenge in reformulating is to establish a direct link between $\theta_S - \theta^*$ and $S$, *without* relying on an inner-loop optimization ($\arg\min$). To achieve this, we can simplify the problem by framing it as an issue of parameter estimation under *data reduction*: Consider the pool $V$ as the full dataset for estimation, while $S$ is its selected subset. The problem then becomes: analyzing how removing a subset $Z$ (where $S = V \setminus Z$) affects the estimated ability.

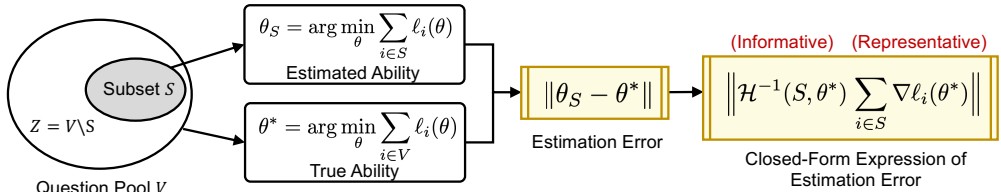

Figure 1: Illustration of subset selection in adaptive testing. The full question pool $V$ is divided into a selected subset $S$ and a remainder $Z$. The estimation error is approximated via first-order (gradient) and second-order (Hessian) terms, capturing $S$'s representativeness and informativeness, respectively.

**Measuring the Impact of Question Reduction on the Ability Estimator.** Obviously, the most direct approach would be to recompute/retrain the parameter estimate from scratch for each choice of $S$, as done in the inner loop's minimization in Eq.(2). But that is computationally prohibitive. Thus, instead of outright removing questions from $V$, we *down-weight* their influence in the ability estimation process. This leads to the definition of a perturbed estimator:

$$\theta_S^\gamma = \arg\min_\theta \frac{1}{|V|} \sum_{i \in V} \ell_i(\theta) - \gamma \sum_{i \in Z} \ell_i(\theta), \tag{4}$$

where $Z$ is the set of down-weighted (or "removed") questions, and $\gamma \in [0, 1/|V|]$. This formulation reduces the contribution of response to $Z$ to the total loss, thereby approximating the effect of excluding them from estimation. For a first-order approximation, we expand the gradient of the loss function evaluated at $\theta_S^\gamma$ using a Taylor expansion of Eq.(4) around $\theta^*$. Since $\theta_S^\gamma$ is a minimizer, its gradient is approximately zero:

$$0 \approx \frac{1}{|V|} \sum_{i \in V} \nabla \ell_i(\theta^*) - \gamma \sum_{i \in Z} \nabla \ell_i(\theta^*) + \left( \frac{1}{|V|} \sum_{i \in V} \nabla^2 \ell_i(\theta^*) - \gamma \sum_{i \in Z} \nabla^2 \ell_i(\theta^*) \right) (\theta_S^\gamma - \theta^*). \tag{5}$$

In particular, when we set $Z = V \setminus S$ and choose $\gamma = 1/|V|$, the perturbed estimator exactly recovers the ability estimate based on the subset $S$, i.e., $\theta_S^\gamma = \theta_S$. Since $\theta^*$ satisfies the optimality condition $\sum_{i \in V} \nabla \ell_i(\theta^*) = 0$, we obtain:

$$\theta_S^\gamma - \theta^* = \theta_S - \theta^* \approx -\mathcal{H}^{-1}(S, \theta^*) \sum_{i \in S} \nabla \ell_i(\theta^*), \tag{6}$$

where $\mathcal{H}(S, \theta^*) = \sum_{i \in S} \nabla^2 \ell_i(\theta^*)$ denotes the Hessian of the loss function for ability estimation, and $\mathcal{H}^{-1}$ denotes its inverse. Here, $\mathcal{H}(S, \theta^*)$ is invertible, which holds under standard regularity conditions in IRT [8, 11]. For complex neural network-based models, where computing the exact Hessian is often intractable, we adopt a quasi-Newton approximation (details are provided in Appendix A). This result can be viewed as an extension of influence function theory [24, 25, 26], which originated in statistics in the 1970s. It characterizes how perturbations in the data affect an estimator. Here, we approximate the effect of selecting a subset $S$ without resorting to explicit re-optimization.

**Closed-Form Expression of Estimation Error.** The key takeaway is that the error in ability estimation based on the selected subset $S$ admits a closed-form expression (Figure 1):

**Lemma 1.** *Let the true ability parameter be $\theta^*$. When using IRT for ability estimation, the estimation error based on any subset $S \subseteq V$ can be directly computed as in Eq.(6). This allows for directly optimizing the selection of $S$ to minimize the estimation error without the need to recompute $\theta_S$:*

$$\min_{|S|=T} \|\theta_S - \theta^*\| \Rightarrow \min_{|S|=T} \left\| \underbrace{\mathcal{H}^{-1}(S, \theta^*)}_{\text{second-order}} \overbrace{\sum_{i \in S} \nabla \ell_i(\theta^*)}^{\text{first-order}} \right\|. \tag{7}$$

This reformulated objective directly quantifies the influence of the selected subset $S$ on the estimation error, bypassing the need for re-optimization of $\theta_S$ in previous nested optimizations. The selection of $S$ balances two critical factors: simultaneously managing both bias minimization (first-order stability) and conditioning of the Hessian (second-order stability):

**Factor 1: First-Order Gradient Alignment**. The term $\sum_{i \in S} \nabla \ell_i(\theta^*)$ captures the aggregate first-order (gradient) contribution of the selected questions. If this sum deviates significantly from zero, it introduces directional bias into the estimated parameter $\theta_S$. Intuitively, *the goal is to find a subset whose gradients "agree" with those of the full question pool*. This ensures that the subset is *representative* of the entire pool in terms of gradient information, and does not skew the estimation.

**Factor 2: Second-Order Information Control**. The Hessian inverse, $\mathcal{H}^{-1}(S, \theta^*)$, controls how the subset's curvature information influences estimation stability. The optimal subset must ensure that the Hessian remains well-conditioned while retaining crucial second-order information to stabilize parameter updates. Consider the case of IRT: the expected Hessian can be approximated by the Fisher information $\mathcal{I}$, i.e., $\mathbb{E}[\mathcal{H}(S, \theta)] \approx -\sum_{i \in S} \mathcal{I}_i(\theta) = -\sum_{i \in S} \alpha_i^2 \cdot p_\theta(q_i, 0) \cdot p_\theta(q_i, 1)$. This suggests that it tries to find informative questions with high discrimination ($\alpha$) and maximum response uncertainty, e.g., $p(q_i, 1) \approx 0.5$[2]

Thus, the best subset is both *diverse* and *informative*—minimizing gradient bias while maintaining a stable Hessian structure—leading to efficient and reliable estimation.

## 3.2 Approximate Optimization for Subset Selection

Based on the above reformulated objective, we aim to select a subset $S$ that minimizes the set function: $\min f(S) = \min \left\| \mathcal{H}^{-1}(S, \theta^*) \sum_{i \in S} \nabla \ell_i(\theta^*) \right\|$. This problem is combinatorial and generally NP-hard. Exhaustively searching for the optimal subset is computationally infeasible for large pool due to the exponential number of possible combinations.

Fortunately, we observe that this objective function exhibits a diminishing marginal gain property, which aligns with the concept of submodularity [29]. Submodularity is a useful concept in combinatorial optimization problems that plays a crucial role in designing efficient approximation [30, 31, 32], e.g., greedy algorithm. More precisely, the objective function $f(S)$ is *approximately submodular*, a property referred to as $\epsilon$-submodularity. This property implies that the incremental benefit of adding an element $x$ decreases as the set grows:

**Theorem 1** ($\epsilon$-Submodularity of the Set Function). *Estimating the ability $\theta$ using IRT, the loss function $\ell(\theta)$ is $\mu$-strongly convex. Assume that the gradient norm and Hessian's spectral norm are bounded: $\|\nabla_\theta \ell_i(\theta)\| \leq G$ and $\|\nabla_\theta^2 \ell_i(\theta)\| \leq H$. The objective $f(S) = \left\| \mathcal{H}^{-1}(S, \theta^*) \sum_{i \in S} \nabla \ell_i(\theta^*) \right\|$ is $\epsilon$-submodular, and $\epsilon = \frac{2G(\mu+H)}{\mu^2 |A|} + \frac{2HG}{\mu^2 |A|^2}$, i.e., for any subsets $A \subseteq B \subseteq V$:*

$$f(A \cup \{x\}) - f(A) \geq f(B \cup \{x\}) - f(B) - \left( \frac{2G(\mu+H)}{\mu^2 |A|} + \frac{2HG}{\mu^2 |A|^2} \right). \tag{8}$$

The proofs can be found in Appendix B. The $\epsilon = \frac{2G(\mu+H)}{\mu^2 |A|} + \frac{2HG}{\mu^2 |A|^2}$ bound decreases as $|A|$ increases. This means the function becomes more submodular as the subset grows, which is intuitive—the marginal benefit of adding a new question becomes more stable as more questions are already selected. This bound provides theoretical justification for using greedy methods: if $\epsilon$ is small (e.g., due to large $|A|$), greedy selection will be near-optimal even though the function is not strictly submodular.

**Greedy Question Selection.** Given that the objective $f(S)$ is $\epsilon$-submodular, we can use a greedy algorithm to iteratively construct an optimal subset $S$. The approximate submodularity property ensures that a greedy selection achieves a near-optimal solution with theoretically bounded suboptimality.

For size-constrained minimization of $f(S)$, a simple reverse greedy algorithm can be adopted. It sequentially selects elements that yield the smallest marginal increase in $f(S)$. Specifically, it starts with an empty subset $S_0 = \emptyset$. At each step $t$, the question that minimizes the marginal gain is selected, formally given by $q_t = \arg\min_{q \in V \setminus S_{t-1}} (f(S_{t-1} \cup \{(q, y)\}) - f(S_{t-1}))$. After selecting $q_t$, the subset is updated as: $S_t = S_{t-1} \cup \{(q_t, y_t)\}$.

In practice, the parameter $\theta^*$ is unknown and we use the estimate $\theta^t$ obtained from $S_t$. The objective function can be approximated: $f(S \mid \theta^t) = \left\| \mathcal{H}^{-1}(S, \theta^t) \sum_{i \in S} \nabla \ell_i(\theta^t) \right\|$. Meanwhile, the true

---

[2]From the perspective of Active Learning, samples near the decision boundary—where the model is most uncertain—are typically the most informative [27, 28].

labels $y$ are also unobserved, we take the expectation over $y$, selecting the next question:

$$q_t = \underset{q \in V \setminus S_{t-1}}{\arg\min} \ \mathbb{E}_y\big[f(S_{t-1} \cup \{q, y\} \mid \theta^{t-1})\big]. \tag{9}$$

The sequential selection process continues until the selected questions reach a predefined maximum size $T$, corresponding to the termination condition of the test. Based on the asymptotic unbiasedness of MLE, we find an upper bound on the approximation error when substituting $\theta^t$ for $\theta^*$.

**Lemma 2** (Approximation Error). *When using IRT for ability estimation, the function $f(S)$ is Lipschitz continuous w.r.t. $\theta$. With probability at least $1 - \delta$, the approximation error incurred by using $\theta^t$ satisfies the upper bound: $|f(S \mid \theta^t) - f(S)| \le \left(\frac{H}{\mu_1} + \frac{MG}{\mu_1 \mu_2}\right) \frac{C(\delta)}{\sqrt{|S_t|}}$, where $\mu_1$, $\mu_2$, $M$, $H$, $G$, and $C$ are model-dependent constants characterizing the properties of the objective function.*

The proofs can be found in Appendix C. The substitution of $\theta^*$ with $\theta^t$ is justified due to the consistency and asymptotic normality of estimators. According to this bounded approximation error in Lemma 2, the error introduced by estimating $\theta^*$ diminishes at a rate of $O(|S_t|^{-1/2})$, ensuring the robustness of the adaptive selection process.

### 3.3 Bias Correction in Ability Estimation: Guessing and Slipping

The idealized ability estimation above assumes that an examinee's responses accurately reflect their true ability. However, in practical testing, the observed response $y$ may not correspond perfectly to true ability due to guessing and slipping [1]. 1) Guessing: An examinee correctly answers a question they should not have been able to solve purely by chance. For example, if a multiple-choice question has three options, random guessing yields a 33.3% success probability; 2) Slipping: An examinee fails to answer a question correctly despite having the ability to do so. It arises due to carelessness, misreading, or other lapses.

Both factors induce *label flipping* in the observed responses $y \in \{0, 1\}$, leading to biased ability estimates $\theta_S$ that contain unpredictable noise. This distortion can be explicitly quantified within this CFAT framework: Consider a response $(q_m, y_m)$ affected by label flipping, resulting in the incorrect label $(q_m, 1 - y_m)$. The corresponding loss becomes: $\widetilde{\ell}_m(\theta) = -(1 - y_m) \log p_\theta(q_m, 1) - y_m \log p_\theta(q_m, 0)$. After incorporating the flipped response, the new ability estimate on $S$ is: $\theta_{S(m)}^\gamma = \arg\min_\theta \frac{1}{|S|} \sum_{i \in S} \ell_i(\theta) - \gamma \ell_m(\theta) + \gamma \widetilde{\ell}_m(\theta)$. Note that this also applies a weighted adjustment rather than physically replacing the affected data. When $\gamma = \frac{1}{|S|}$, the correction becomes equivalent to a full replacement of the original response.

Applying a Taylor expansion around $\theta_S$, similar to the derivations in Section 3.1, we approximate:

$$\theta_{S(m)} = \theta_S + \big[\mathcal{H}(S \setminus q_m, \theta_S) + \widetilde{\mathcal{H}}(q_m, \theta_S)\big]^{-1} (1 - 2y_m) \nabla \log \frac{p_\theta(q_m, 1)}{p_\theta(q_m, 0)}, \tag{10}$$

where $\widetilde{\mathcal{H}}(q_m, \theta_S) = \nabla^2 \widetilde{\ell}_m(\theta_S)$. The term $\Delta\theta_{S(m)} = \theta_{S(m)} - \theta_S$ *provides a quantitative measure of how a flipped response skews the estimate*. Even if we cannot pinpoint the specific flipped samples, analyzing the expected effect enables us to understand the direction and magnitude of the systematic bias caused by response errors. See Appendix D for a detailed derivation of Eq.(10).

Thus, instead of relying on the potentially biased estimator $\theta_S$, we introduce a bias-corrected ability estimate by subtracting the expected distortion: $\theta_S - \mathbb{E}[\Delta\theta] = \theta_S - \sum_{m \in S} \big[\pi_g(1 - y_m) + \pi_s y_m\big] \Delta\theta_{S(m)}$, where $\pi_g$ is the guessing probability, capturing the likelihood of obtaining a correct response by chance, and $\pi_s$ is the slipping probability, representing the likelihood of incorrect responses despite having the requisite ability. The complete CFAT framework is shown in Algorithm 1

## 4 Experiments

**Evaluation Tasks.** To assess the efficiency of question selection algorithms in adaptive testing, we evaluate the accuracy of ability estimation under a fixed test length. Specifically, we compare the final estimated ability $\theta_S$, where $S$ represents the selected question subset chosen by different selection algorithms. The evaluation is conducted across two primary tasks [9, 7]: 1) Student Performance

---
**Algorithm 1:** The proposed framework CFAT
---
**Require:** $V$ - Question pool, $p_\theta$ - Parameterized probability model (IRT or neural network), $\pi_g$ - Guessing probability, $\pi_s$ - Slipping probability

**Initialize:** Initialize the ability estimate $\theta^0$ and responses data $S_0 \leftarrow \emptyset$.

1 **for** $t = 1$ **to** $T$ **do**
2 $\quad$ Select the next question $q_t$ by minimizing the set function:
$\quad\quad q_t = \arg\min_{q \in V \setminus S_{t-1}} \mathbb{E}_y f(S_{t-1} \cup \{q, y\} \mid \theta^{t-1})$.
3 $\quad$ Obtain the examinee's response label $y_t$: $S_t \leftarrow S_{t-1} \cup \{(q_t, y_t)\}$.
4 $\quad$ Update examinee's ability estimate: $\quad \theta^t \leftarrow \arg\min_{\theta \in \Theta} \sum_{i \in S_t} \ell_i(\theta)$.
5 $\quad$ Apply bias correction to adjust for response errors:
$\quad\quad \theta^t \leftarrow \theta^t - \sum_{m \in S_t} \left[ \pi_g(1 - y_m) + \pi_s y_m \right] \Delta\theta_{S_t(m)}$

**Output:** The examinee's ability estimate $\theta_S = \theta^T$ using the responses on the selected $S$.
---

Prediction: Using the estimated $\theta_S$, predicting students' responses (correct/incorrect) on a held-out test set and measure predictive performance using Accuracy and AUC; 2) Ability Estimation Error: In a simulation setting, the ground-truth ability $\theta^*$ is constructed and simulate students' response behavior during testing. We then compute the estimation error using the Mean Squared Error (MSE) $\mathbb{E}||\theta_S - \theta^*||^2$.

**Experimental Implementation Details.** We set the maximum test length to $|S| = T = 20$, consistent with typical adaptive tests. All methods are implemented in PyTorch and trained on a Tesla V100 GPU. Hyperparameters are tuned via grid search, with batch size 64, learning rate 0.001, and behavioral noise parameters $\pi_g = 0.002$, $\pi_s = 0.001$. Optimization is performed using Adam.

Following [9, 1], we split examinees into 70% training, 20% validation, and 10% testing. The training set is used to estimate question parameters and train some data-driven models. During validation and testing, we simulate adaptive testing: Specifically, for the student performance prediction task, each examinee's responses are divided into a candidate set $V_i$ (for question selection and ability estimation) and a meta set $M_i$ (for evaluation via Accuracy/AUC). At each step, a question is selected from $V_i$, ability is updated, and performance is evaluated on $M_i$. For ability estimation error, ground-truth abilities $\theta^*$ are estimated from full responses, allowing simulated examinees to answer any question in $V$ for direct error computation.

**Datasets.** We conduct experiments on three widely used educational testing benchmark datasets: ASSIST, NIPS-EDU, and EXAM: ASSIST [33] is derived from the online educational platform ASSISTments and contains examinees' practice logs on mathematics; NeurIPS-EDU [34] originates from the NeurIPS 2020 Education Challenge, comprising a large-scale dataset collected from examinees' responses to questions on Eedi, an educational platform. EXAM is a dataset from iFLYTEK Co., Ltd. that records junior high school students' performances on mathematical exams. The implementation code is available on: `https://github.com/54zy/CFAT`.

**Compared Approaches.** For ability estimation, we consider both classical IRT model and neural network-based approaches: NeuralCDM [35], a flexible framework that generalizes various IRT and cognitive diagnosis models, e.g., MIRT [36] and Matrix Factorization [37, 38]. The objective of our experiments is to compare the proposed selection algorithm against existing selection methods in terms of their impact on ability estimation. Thus, we evaluate the following SOTA algorithms as baselines: **Random Selection** serves as a benchmark by selecting questions uniformly at random, providing a reference for assessing the improvements achieved by other algorithms; **Fisher Information** [14] and **KL Information** [19] are classical methods that prioritize questions based on their informativeness; **MAAT** [23] uses active learning to balance uncertainty and diversity. **BOBCAT** [9] and **UATS** [8] apply meta-learning to solve the nested selection problem via cross-entropy minimization. **NCAT** [10] and **GMOCAT** [2] frame selection as reinforcement learning, leveraging transformers and GNNs to train a data-driven selection policy. **BECAT** [7] uses a greedy heuristic based on first-order gradient approximation, between the selected subset and the entire question pool.

Table 1: The performances on Student Performance Prediction. It reports ACC and AUC at 5th, 10th, and 20th step (subset size). Panel 1 presents results based on the IRT model for ability estimation, while Panel 2 uses a neural network-based model (NeuralCDM). Note that information/uncertainty-based methods (e.g., Fisher) are not applicable to deep models. Bold values indicate statistically significant improvements (p-value $< 0.01$) over the best baseline.

| Method | ASSIST (ACC/AUC) | | | NIPS-EDU (ACC/AUC) | | | EXAM (ACC/AUC) | | |
|---|---|---|---|---|---|---|---|---|---|
| | @5 | @10 | @20 | @5 | @10 | @20 | @5 | @10 | @20 |
| Random | 70.89/70.78 | 71.99/71.84 | 73.02/72.45 | 66.57/69.02 | 68.11/71.42 | 70.00/73.90 | 77.58/70.34 | 77.22/71.83 | 80.33/74.09 |
| Fisher | 71.87/71.22 | 72.63/72.30 | 73.11/73.56 | 67.70/70.62 | 70.59/73.51 | 71.23/76.33 | 77.35/70.51 | 79.75/72.25 | 83.03/75.90 |
| KL | 71.95/71.31 | 72.68/72.50 | 73.13/73.57 | 67.09/69.71 | 69.29/73.30 | 70.41/75.73 | 77.37/70.58 | 79.22/72.11 | 83.01/75.73 |
| MAAT | 72.11/71.24 | 72.03/72.38 | 73.20/73.05 | 66.44/69.31 | 69.10/71.12 | 69.27/73.40 | 75.27/70.32 | 77.99/72.12 | 80.12/73.67 |
| BOBCAT | 72.33/71.72 | 72.56/72.18 | 73.78/73.31 | 69.55/74.41 | 70.99/75.66 | 71.71/76.44 | 80.61/68.29 | 83.81/72.02 | 83.44/72.82 |
| NCAT | 72.22/71.66 | 72.52/72.38 | 73.83/73.51 | 67.30/72.11 | 70.68/75.80 | 71.91/76.66 | 80.92/70.72 | 83.96/72.67 | 83.88/74.19 |
| UATS | 72.29/72.82 | 72.04/72.74 | 74.14/74.84 | 67.58/73.33 | 70.50/74.82 | 71.84/76.57 | 79.17/70.22 | 82.33/73.29 | 84.91/75.24 |
| BECAT | 71.92/71.34 | 73.01/72.73 | 73.96/73.63 | 66.98/73.15 | 71.61/75.85 | 72.00/76.82 | 80.93/70.74 | 83.80/72.88 | 84.20/75.03 |
| **CFAT** | **72.86/73.48** | **73.37/73.26** | **74.29/75.22** | **69.62/74.55** | **72.25/76.22** | **73.87/78.03** | **81.11/71.03** | **84.13/73.80** | **86.05/77.83** |

| Method | ASSIST (ACC/AUC) | | | NIPS-EDU (ACC/AUC) | | | EXAM (ACC/AUC) | | |
|---|---|---|---|---|---|---|---|---|---|
| | @5 | @10 | @20 | @5 | @10 | @20 | @5 | @10 | @20 |
| Random | 71.21/71.02 | 72.53/72.08 | 72.51/72.83 | 67.13/69.39 | 68.42/71.51 | 70.59/74.93 | 79.80/72.48 | 78.33/74.52 | 79.31/78.22 |
| MAAT | 72.09/70.74 | 72.31/72.03 | 71.75/72.29 | 67.83/70.00 | 70.42/72.58 | 70.63/75.85 | 82.87/70.22 | 82.55/74.29 | 83.72/79.36 |
| BOBCAT | 72.64/71.46 | 72.77/72.73 | 73.80/72.82 | 71.02/76.12 | 72.46/77.82 | 73.42/79.06 | 78.13/78.28 | 78.13/81.45 | 78.04/79.53 |
| NCAT | 72.29/71.64 | 72.62/72.34 | 73.92/73.56 | 70.43/74.12 | 72.84/77.92 | 73.44/79.09 | 82.33/78.54 | 83.13/81.46 | 81.44/79.35 |
| UATS | 73.02/72.32 | 72.92/73.05 | 73.16/72.73 | **71.87**/75.13 | 73.13/78.12 | 74.14/79.70 | 81.26/77.12 | 82.46/80.92 | 83.79/80.82 |
| BECAT | 72.30/71.61 | 73.11/72.87 | 74.13/73.70 | 71.33/76.31 | 73.07/78.24 | 73.58/79.26 | 82.84/78.75 | 83.22/81.49 | 84.77/79.70 |
| **CFAT** | **74.13/72.92** | **73.45/73.98** | **74.53/74.38** | 71.20/**76.19** | **74.43/ 78.38** | **74.72/81.77** | **83.33/80.98** | **84.12/82.87** | **85.12/81.66** |

## 4.1 Experimental Results

We evaluate the proposed CFAT framework on two core tasks (i.e., student performance prediction and ability estimation error) across three benchmark datasets.

**Task 1: Student Performance Prediction:** This task assesses the efficiency of ability estimation in adaptive testing. Specifically, we compare the prediction accuracy of response labels (correct/incorrect) under different question selection strategies, where each algorithm selects the same number of questions. As shown in Table 1, we report the AUC and ACC scores for each method at the 5th, 10th, and 20th steps. The proposed CFAT consistently achieves the highest prediction accuracy under limited question settings. Notably, simple greedy algorithm CFAT outperforms neural network methods, e.g., reinforcement learning (NCAT) and meta-learning approaches (BOBCAT, UATS), by an average margin of 2% in AUC.

These results support our central claim: formulating the subset selection problem with a closed-form objective yields better performance than complex nested paradigm. Furthermore, CFAT outperforms the gradient-based BECAT. It highlights the advantage of incorporating second-order information (i.e., the Hessian matrix) over relying solely on first-order gradients. This superiority is observed both theoretically and empirically at scale. Although our theoretical derivation is grounded in the classical IRT model, the CFAT framework also demonstrates strong performance when applied to neural network ability estimation models (NeuralCDM). This suggests that our subset selection formulation and its approximation are generalizable and extensible across different modeling paradigms.

**Task 2: Ability Estimation Error:** To evaluate the accuracy of ability estimation, we adopt a widely used simulation protocol in adaptive testing. Specifically, we treat the ability estimate derived from an examinee's full response data, denoted as $\theta^*$, as the ground truth. During the testing process, this ground-truth ability allows us to simulate response labels for any question, while the tested algorithms only have access to observed response data and not the true ability. Figure 2 illustrates the estimation error $\|\theta^t - \theta^*\|$ over the testing process, where $\theta^t$ denotes the estimated ability at step $t$. The results show that our proposed CFAT achieves comparable estimation accuracy using only 30%–45% of the questions required by random selection. Compared to recent SOTA methods (e.g., UATS), CFAT reduces the number of required questions by at least 15%.

Although CFAT exhibits a relatively slower start in the early stages of testing, its estimation error decreases rapidly as more questions are selected. This initial lag is somewhat less favorable compared to other data-driven methods (e.g., meta learning) that are good at mitigating the cold-start problem [39]. These empirical observations align well with the analysis in Theorem 1, which demonstrates that the submodular nature guarantees the near-optimality of the greedy question selection algorithm as the selected subset grows.

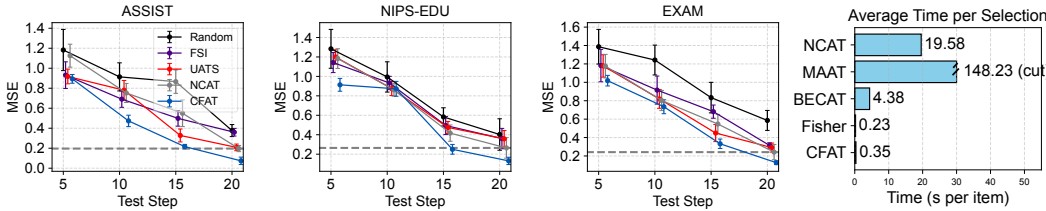

Figure 2: (a) Simulation results for ability estimation: MSE of ability estimation, $\mathbb{E}\|\theta^t - \theta_0\|^2$, under different subset sizes (steps) for five representative question selection algorithms. Results are averaged over 10 repetitions, with error bars indicating the standard deviation. (b) Average time required to select a single question for each method.

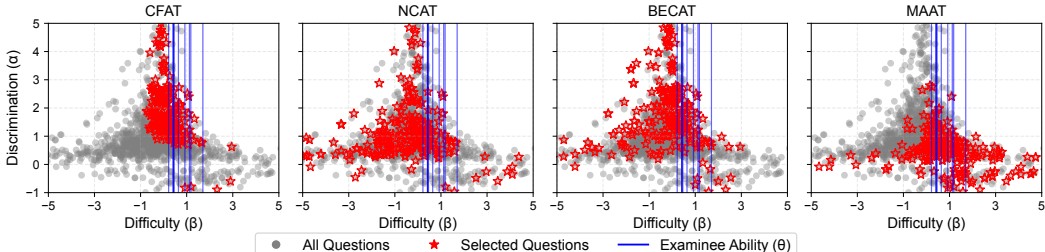

Figure 3: Characteristics of selected questions across different methods. We randomly sample 10 examinees and compare the question subsets selected by CFAT with several SOTA baselines. The distributions of question difficulty and discrimination parameters are visualized.

**Analysis of Computational Efficiency and Subset Characteristics.** We compare the computational efficiency of different question selection algorithms to assess their practical applicability in large-scale testing (note: no acceleration techniques or engineering optimizations are applied). Specifically, in Figure 2(b), we report the average time required to select a single question. CFAT demonstrates higher efficiency compared to SOTA methods e.g., MAAT and BECAT–achieving approximately 12× speedup over BECAT. Notably, CFAT matches the speed of the classical Fisher information method, while simultaneously delivering at least a 20% improvement in estimation accuracy, as evidenced in Figure 2(a). Meanwhile, Figure 3 illustrates the characteristics of question subsets selected by different methods, along with the true ability estimates $\theta^*$ of 10 randomly sampled examinees. As shown, the questions selected by CFAT tend to have higher discrimination and are well-aligned with the examinees' ability levels (i.e., question difficulty closely matches ability). In contrast, other methods often prioritize diverse questions, many of which are "outliers"–either too easy or too difficult for the examinees. Such low-discrimination or mismatched questions tend to be less informative and may hinder accurate ability estimation [40].

**Reliability under Guessing and Slipping Noise.**
In real-world scenarios, examinee's responses may be affected by guessing (label flipping $0 \rightarrow 1$) or slipping (label flipping $1 \rightarrow 0$)[41]. To model this, label noise is introduced in above simulation by flipping response labels with a certain probability. Table 2 illustrates the estimation error across different algorithms as the flipping probability increases. Previous approaches exhibit significant performance degradation under noise. In contrast, CFAT consistently maintains lower estimation error, outperforming its ablated version (CFAT w/o correction), which lacks the bias correction term. Notably, under high noise levels (e.g., 20% label flipping), CFAT still achieves stable and accurate ability estimates. These results empirically validate our theoretical analysis in Section 3.3, highlighting the effectiveness of incorporating a correction term when estimating the ability $\theta_S$.

Table 2: MSE for different selection algorithms in ASSIST under varying levels of label perturbation (Step=20). Perturbation is applied to the examinee's response label. 'No Pert.' denotes MSE without any label noise.

| Method | No Pert. | 5% Pert. | 10% Pert. | 20% Pert. |
|---|---|---|---|---|
| Random | 0.3765 | 0.3827 (+0.0062) | 0.4936 (+0.1171) | 0.6681 (+0.2316) |
| KL | 0.3599 | 0.3638 (+0.0039) | 0.4744 (+0.1145) | 0.5869 (+0.2270) |
| BECAT | 0.3697 | 0.3741 (+0.0044) | 0.4814 (+0.1117) | 0.6005 (+0.2308) |
| GMOCAT | 0.2322 | 0.2375 (+0.0053) | 0.2956 (+0.0634) | 0.3478 (+0.1156) |
| **CFAT** (w/o correction) | 0.1962 | 0.2024 (+0.0062) | 0.3121 (+0.1159) | 0.4324 (+0.2362) |
| **CFAT** | **0.1738** | **0.1778 (+0.0040)** | **0.2082 (+0.0344)** | **0.2770 (+0.1032)** |

# 5 Conclusion

This paper addresses the subset selection problem in ability estimation: how to select a small question subset such that the estimated ability closely approximates the true ability. Instead of relying on the traditional nested optimization paradigm, we derive a closed-form objective that allows for direct optimization. It shows that a simple greedy algorithm can effectively solve this problem, and partially correct the bias of the ability estimator. Extensive experiments demonstrate that it is computationally efficient, yields more accurate ability estimates, better adapts to individuals, and remains robust under high-noise conditions.

## Acknowledgements

This research was supported by grants from the National Key Research and Development Program of China (Grant No. 2024YFC3308200, 62477044), the National Natural Science Foundation of China (62525606), the Key Technologies R & D Program of Anhui Province (No. 202423k09020039), the National Education Science Planning Project (Grant No. ZSA240466), and the Fundamental Research Funds for the Central Universities.

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

# A  Complete Algorithmic Procedure of CFAT

This section presents the complete optimization process of CFAT in practical adaptive testing. Algorithm 2 provides a detailed illustration of the gradient-based ability estimation procedure. However, during the actual question selection phase of CFAT, computing the inverse of the Hessian matrix is required. While this is tractable for traditional IRT models, it becomes computationally infeasible for neural network-based models due to the high dimensionality and complexity of their parameter spaces. To address this, Algorithm 3 introduces an efficient approximation of the Hessian inverse using a quasi-Newton method [42]. This enables the practical deployment of CFAT in neural network settings, and forms the basis of our complete CFAT algorithm tailored for deep learning adaptive testing systems.

---

**Algorithm 2:** Full Procedure of CFAT

---

**Require:** $V$ - Question pool, $p_\theta$ - Parameterized probability model (IRT or neural network), $\pi_g$ - Guessing probability, $\pi_s$ - Slipping probability.

**Initialize:** Initialize the ability estimate $\theta^0$ and responses data $S_0 \leftarrow \emptyset$.

1 **for** $t = 1$ **to** $T$ **do**

2     Select the next question $q_t$ by minimizing the set function:
    $q_t = \arg\min_{q \in V \setminus S_{t-1}} \mathbb{E}_y f(S_{t-1} \cup \{q, y\} \mid \theta^{t-1})$.

3     Obtain the examinee's response label $y_t$: $S_t \leftarrow S_{t-1} \cup \{(q_t, y_t)\}$.

4     Initialize examinee's ability estimate $\theta_0^t \leftarrow \theta_K^{t-1}$.

5     Update examinee's ability estimate:

6     **for** $k = 1$ **to** $K$ **do**

7        Update $\theta_k^t$: $\theta_k^t \leftarrow \theta_{k-1}^t - \alpha \nabla \ell_i(\theta_{k-1}^t)$.

8     Apply bias correction to adjust for response errors:
    $\theta_K^t \leftarrow \theta_K^t - \sum_{m \in S_t}[\pi_g(1 - y_m) + \pi_s y_m]\Delta\theta_{S_t}(m)$

**Output:** The examinee's ability estimate $\theta_S = \theta_K^T$ using the responses on the selected $S$.

---

**Algorithm 3:** Full Procedure of CFAT (Approximate)

---

**Require:** $V$ - Question pool, $p_\theta$ - Parameterized probability model (IRT or neural network), $\pi_g$ - Guessing probability, $\pi_s$ - Slipping probability, $\alpha$ - learning rate.

**Initialize:** Initialize the ability estimate $\theta^0$ and responses data $S_0 \leftarrow \emptyset$, the approximation of the inverse of Hessian matrix $\mathcal{H}_K^{-1^{(0)}} \leftarrow I$ and the examinee's ability estimate $\theta_K^0$.

1 **for** $t = 1$ **to** $T$ **do**

2     Let $\mathcal{H}^{-1} \leftarrow \mathcal{H}_K^{-1^{(t-1)}}$ and select the next question $q_t$ by minimizing the set function:
    $q_t = \arg\min_{q \in V \setminus S_{t-1}} \mathbb{E}_y f(S_{t-1} \cup \{q, y\} \mid \theta^{t-1})$.

3     Obtain the examinee's response label $y_t$: $S_t \leftarrow S_{t-1} \cup \{(q_t, y_t)\}$.

4     Initialize examinee's ability estimate $\theta_0^t \leftarrow \theta_K^{t-1}$.

5     Update examinee's ability estimate:

6     **for** $k = 1$ **to** $K$ **do**

7        Calculate the search direction: $d_k \leftarrow -\mathcal{H}_{k-1}^{-1^{(t)}} \sum_{i \in S} \nabla \ell_i(\theta_{k-1}^t)$.

8        Update $\theta_k^t$: $\theta_k^t \leftarrow \theta_{k-1}^t + \alpha d_k$.

9        Let $u_k \leftarrow \theta_k^t - \theta_{k-1}^t$ and $v_k \leftarrow \sum_{i \in S} \nabla \ell_i(\theta_k^t) - \sum_{i \in S} \nabla \ell_i(\theta_{k-1}^t)$.

10        Update the approximation of the inverse of Hessian matrix $H$:

$$\mathcal{H}_k^{-1^{(t)}} \leftarrow \mathcal{H}_{k-1}^{-1^{(t)}} + \frac{u_k u_k^{\mathrm{T}}}{u_k^{\mathrm{T}} v_k} - \frac{\mathcal{H}_{k-1}^{-1^{(t)}} v_k v_k^{\mathrm{T}} \mathcal{H}_{k-1}^{-1^{(t)}}}{v_k^{\mathrm{T}} \mathcal{H}_{k-1}^{-1^{(t)}} v_k}$$

11     Apply bias correction to adjust for response errors:
    $\theta_K^t \leftarrow \theta_K^t - \sum_{m \in S_t}[\pi_g(1 - y_m) + \pi_s y_m]\Delta\theta_{S_t}(m)$

**Output:** The examinee's ability estimate $\theta_S = \theta_K^T$ using the responses on the selected $S$.

---

# B Proofs of Theorem 1

**Theorem 1** ($\epsilon$-Submodularity of the Set Function). *Estimating the ability parameter $\theta$ using IRT, the loss function $\ell(\theta)$ is assumed to be $\mu$-strongly convex [7]. Assume that the gradient norm and Hessian's spectral norm are bounded by $\|\nabla_\theta \ell_i(\theta)\| \leq G$ and $\|\nabla_\theta^2 \ell_i(\theta)\| \leq H$. The subset selection objective $f(S) = \left\| \mathcal{H}^{-1}(S, \theta^*) \sum_{i \in S} \nabla \ell_i(\theta^*) \right\|$ is $\epsilon$-submodular, and $\epsilon = \frac{2G(\mu+H)}{\mu^2|A|} + \frac{2HG}{\mu^2|A|^2}$, i.e., for any subsets $A \subseteq B \subseteq V$:*

$$f(A \cup \{x\}) - f(A) \geq f(B \cup \{x\}) - f(B) - \left( \frac{2G(\mu+H)}{\mu^2|A|} + \frac{2HG}{\mu^2|A|^2} \right). \tag{11}$$

*Proof.* Based on Lemma 1, the objective function can be formulated as:

$$f(S) = \left\| \mathcal{H}^{-1}(S, \theta^*) \sum_{i \in S} \nabla \ell_i(\theta^*) \right\|,$$

where $\mathcal{H}(S, \theta) = \sum_{i \in S} \nabla^2 \ell_i(\theta)$ is Hessian matrix.

We assume the following boundedness conditions on the loss function $\ell_i(\theta)$ over the set $V$: 1) The gradient norm is upper-bounded: $\|\nabla_\theta l_i(\theta)\| \leq G$; 2) The spectral norm of the Hessian is also bounded: $\|\nabla^2 \ell_i(\theta)\| \leq H$.

For IRT-based ability estimation, the loss function $\ell(\theta)$ is known to be $\mu$-strongly convex [7]. As a result, the Hessian matrix satisfies: $\mathcal{H}(x, \theta) \succeq \mu I_n$. This implies that all eigenvalues of $\mathcal{H}(x, \theta)$ satisfy $\lambda_{\min}(\mathcal{H}(x, \theta)) \geq \mu$. Considering the inverse Hessian matrix $\mathcal{H}(x, \theta)^{-1}$, we have $\lambda(\mathcal{H}(x, \theta)^{-1}) = \frac{1}{\lambda(\mathcal{H}(x,\theta))}$. Thus, the largest eigenvalue of $\mathcal{H}(x, \theta)^{-1}$ satisfies:

$$\lambda_{\max}(\mathcal{H}(x, \theta)^{-1}) = \frac{1}{\lambda_{\min}(\mathcal{H}(x, \theta))} \leq \frac{1}{\mu}. \tag{12}$$

Since the spectral norm (2-norm) of a symmetric matrix is equal to its largest eigenvalue, we conclude:

$$\|\mathcal{H}(x, \theta)^{-1}\| \leq \frac{1}{\mu}. \tag{13}$$

Define: $\mathcal{H}_S = \mathcal{H}(S, \theta^*)$ as the Hessian of the current subset $S$. $g_S = \sum_{i \in S} \nabla \ell_i(\theta^*)$ as the cumulative gradient for the current subset. When we add a new element $x$ to $S$, the function gain is given by:

$$\Delta(x, S) = f(S \cup \{x\}) - f(S) = \left\| (\mathcal{H}_S + \nabla_\theta^2 \ell_x(\theta^*))^{-1}(g_S + \nabla_\theta \ell_x(\theta^*)) \right\| - \left\| \mathcal{H}_S^{-1} g_S \right\|. \tag{14}$$

To prove that the function is $\epsilon$-submodular, we must show that for any subsets $A \subseteq B \subseteq V$, the following inequality holds:

$$\Delta(x, A) \geq \Delta(x, B) - \epsilon, \quad \text{where } \epsilon > 0. \tag{15}$$

For simplicity, we define $\Delta \mathcal{H} = \nabla_\theta^2 \ell_x(\theta^*)$ and $\Delta g = \nabla_\theta \ell_x(\theta^*)$. Now, applying the first-order approximation of the inverse matrix [43]:

$$(\mathcal{H}_S + \Delta \mathcal{H})^{-1} \approx \mathcal{H}_S^{-1} - \mathcal{H}_S^{-1} \Delta \mathcal{H} \mathcal{H}_S^{-1} + O(\|\Delta \mathcal{H}\|^2). \tag{16}$$

Substituting this into Eq.(14) for $\Delta(x, S)$:

$$\Delta(x, S) \approx \left\| \mathcal{H}_S^{-1} g_S + \mathcal{H}_S^{-1} \Delta g - \mathcal{H}_S^{-1} \Delta \mathcal{H} \mathcal{H}_S^{-1} g_S - \mathcal{H}_S^{-1} \Delta \mathcal{H} \mathcal{H}_S^{-1} \Delta g \right\| - \left\| \mathcal{H}_S^{-1} g_S \right\|. \tag{17}$$

Using the triangle inequality, the gain associated with subset $A$ satisfies:

$$\begin{aligned}
\Delta(x, A) &\approx \left\| \mathcal{H}_A^{-1} g_A + \mathcal{H}_A^{-1} \Delta g - \mathcal{H}_A^{-1} \Delta \mathcal{H} \mathcal{H}_A^{-1} g_A - \mathcal{H}_A^{-1} \Delta \mathcal{H} \mathcal{H}_A^{-1} \Delta g \right\| - \left\| \mathcal{H}_A^{-1} g_A \right\| \\
&\geq \left\| \mathcal{H}_A^{-1} g_A + \mathcal{H}_A^{-1} \Delta g \right\| - \left\| \mathcal{H}_A^{-1} \Delta \mathcal{H} \mathcal{H}_A^{-1} g_A + \mathcal{H}_A^{-1} \Delta \mathcal{H} \mathcal{H}_A^{-1} \Delta g \right\| - \left\| \mathcal{H}_A^{-1} g_A \right\| \\
&\geq \left\| \mathcal{H}_A^{-1} g_A \right\| - \left\| \mathcal{H}_A^{-1} \Delta g \right\| - \left\| \mathcal{H}_A^{-1} \Delta \mathcal{H} \mathcal{H}_A^{-1} g_A \right\| - \left\| \mathcal{H}_A^{-1} \Delta \mathcal{H} \mathcal{H}_A^{-1} \Delta g \right\| - \left\| \mathcal{H}_A^{-1} g_A \right\| \\
&= - \left\| \mathcal{H}_A^{-1} \Delta g \right\| - \left\| \mathcal{H}_A^{-1} \Delta \mathcal{H} \mathcal{H}_A^{-1} g_A \right\| - \left\| \mathcal{H}_A^{-1} \Delta \mathcal{H} \mathcal{H}_A^{-1} \Delta g \right\|. \tag{18}
\end{aligned}$$

Using norm bounds, we can further estimate:

$$\Delta(x, A) \geq -\frac{G}{\mu|A|} - \frac{HG}{\mu^2|A|} - \frac{HG}{\mu^2|A|^2}. \tag{19}$$

Similarly, for subset $B$, we obtain:

$$\begin{aligned}
\Delta(x, B) &\approx \left\|\mathcal{H}_B^{-1}g_B + \mathcal{H}_B^{-1}\Delta g - \mathcal{H}_B^{-1}\Delta\mathcal{H}\mathcal{H}_B^{-1}g_B - \mathcal{H}_B^{-1}\Delta\mathcal{H}\mathcal{H}_B^{-1}\Delta g\right\| - \left\|\mathcal{H}_B^{-1}g_B\right\| \\
&\leq \left\|\mathcal{H}_B^{-1}g_B\right\| + \left\|\mathcal{H}_B^{-1}\Delta g\right\| + \left\|\mathcal{H}_B^{-1}\Delta\mathcal{H}\mathcal{H}_B^{-1}g_B\right\| + \left\|\mathcal{H}_B^{-1}\Delta\mathcal{H}\mathcal{H}_B^{-1}\Delta g\right\| - \left\|\mathcal{H}_B^{-1}g_B\right\| \\
&\leq \frac{G}{\mu|B|} + \frac{HG}{\mu^2|B|} + \frac{HG}{\mu^2|B|^2}. \tag{20}
\end{aligned}$$

Since $A \subseteq B$, the difference:

$$\Delta(x, A) - \Delta(x, B) \geq -\frac{2G(\mu + H)}{\mu^2|A|} - \frac{2HG}{\mu^2|A|^2}. \tag{21}$$

Thus, the parameter $\epsilon = \frac{2G(\mu+H)}{\mu^2|A|} + \frac{2HG}{\mu^2|A|^2}$. This completes the proof. $\qquad\square$

## C   Proofs of Lemma 2

**Lemma 2.** *When using IRT for ability estimation, the function $f(S)$ is Lipschitz continuous with respect to $\theta$. Furthermore, with probability at least $1 - \delta$, the approximation error incurred by using $\theta^t$ satisfies the upper bound: $|f(S \mid \theta^t) - f(S)| \leq \left(\frac{H}{\mu_1} + \frac{MG}{\mu_1\mu_2}\right)\frac{C(\delta)}{\sqrt{|S_t|}}$, where $\mu_1$, $\mu_2$, $M$, $H$, $G$, and $C$ are model-dependent constants characterizing the properties of the objective function.*

*Proof.* We first should prove that $f(S \mid \theta) = \left\|\left(\sum_{i \in S}\nabla_\theta^2\ell_i(\theta)\right)^{-1}\sum_{i \in S}\nabla_\theta\ell_i(\theta)\right\|$ is Lipschitz continuous with respect to $\theta$, we analyze its sensitivity to small changes in $\theta$.

Since the gradient $\nabla_\theta\ell_i(\theta)$ is continuously differentiable in $\theta$, the Mean Value Theorem guarantees the existence of some $\xi_i$ between $\theta_1$ and $\theta_2$ such that

$$\nabla_\theta\ell_i(\theta_1) - \nabla_\theta\ell_i(\theta_2) = \nabla_\theta^2\ell_i(\xi_i)(\theta_1 - \theta_2). \tag{22}$$

Assuming that $\|\nabla_\theta^2\ell_i(\theta)\| \leq H$ and taking the norm and summing over $i \in S$ gives

$$\left\|\sum_{i \in S}\nabla_\theta\ell_i(\theta_1) - \sum_{i \in S}\nabla_\theta\ell_i(\theta_2)\right\| \leq \sum_{i \in S}\left\|\nabla_\theta^2\ell_i(\xi_i)\right\|\|\theta_1 - \theta_2\| \leq H|S|\|\theta_1 - \theta_2\|. \tag{23}$$

Similarly, assuming $\|\nabla_\theta^3\ell_i(\theta)\| \leq M$, there exists some $\eta_i$ between $\theta_1$ and $\theta_2$ such that

$$\left\|\sum_{i \in S}\nabla_\theta^2\ell_i(\theta_1) - \sum_{i \in S}\nabla_\theta^2\ell_i(\theta_2)\right\| \leq \sum_{i \in S}\left\|\nabla_\theta^3\ell_i(\eta_i)\right\|\|\theta_1 - \theta_2\| \leq M|S|\|\theta_1 - \theta_2\|. \tag{24}$$

Define: $\mathcal{H}(\theta) = \sum_{i \in S}\nabla_\theta^2\ell_i(\theta)$ and $g(\theta) = \sum_{i \in S}\nabla_\theta\ell_i(\theta)$. We analyze

$$\begin{aligned}
|f(S, \theta_1) - f(S, \theta_2)| &= \left|\|\mathcal{H}(\theta_1)^{-1}g(\theta_1)\| - \|\mathcal{H}(\theta_2)^{-1}g(\theta_2)\|\right| \\
&\leq \|\mathcal{H}(\theta_1)^{-1}g(\theta_1) - \mathcal{H}(\theta_2)^{-1}g(\theta_2)\| \\
&= \|\mathcal{H}(\theta_1)^{-1}g(\theta_1) - \mathcal{H}(\theta_1)^{-1}g(\theta_2) + \mathcal{H}(\theta_1)^{-1}g(\theta_2) - \mathcal{H}(\theta_2)^{-1}g(\theta_2)\| \\
&\leq \|\mathcal{H}(\theta_1)^{-1}(g(\theta_1) - g(\theta_2))\| + \|(\mathcal{H}(\theta_1)^{-1} - \mathcal{H}(\theta_2)^{-1})g(\theta_2)\|. \tag{25}
\end{aligned}$$

For the first term, using matrix norm properties:

$$\|\mathcal{H}(\theta_1)^{-1}(g(\theta_1) - g(\theta_2))\| \leq \|\mathcal{H}(\theta_1)^{-1}\| \cdot \|g(\theta_1) - g(\theta_2)\|. \tag{26}$$

Assuming $\|\mathcal{H}(\theta_1)^{-1}\| \leq \frac{1}{|S|\mu_1}$ in a well-conditioned region (similar to Theorem 1), we obtain:

$$\|\mathcal{H}(\theta_1)^{-1}(g(\theta_1) - g(\theta_2))\| \leq \frac{H}{\mu_1}\|\theta_1 - \theta_2\|. \tag{27}$$

For the second term, using:

$$\mathcal{H}(\theta_1)^{-1} - \mathcal{H}(\theta_2)^{-1} = \mathcal{H}(\theta_1)^{-1}(\mathcal{H}(\theta_2) - \mathcal{H}(\theta_1))\mathcal{H}(\theta_2)^{-1},$$

and assuming $\|\mathcal{H}(\theta_2)^{-1}\| \leq \frac{1}{|S|\mu_2}$, we obtain:

$$\|(\mathcal{H}(\theta_1)^{-1} - \mathcal{H}(\theta_2)^{-1})g(\theta_2)\| \leq \|\mathcal{H}(\theta_1)^{-1}\|\|\mathcal{H}(\theta_2) - \mathcal{H}(\theta_1)\|\|\mathcal{H}(\theta_2)^{-1}\|\|g(\theta_2)\|. \tag{28}$$

Using our previous bound $\|\mathcal{H}(\theta_2) - \mathcal{H}(\theta_1)\| \leq M|S|\|\theta_1 - \theta_2\|$ and assuming $\|g(\theta)\| \leq |S|G$ in a bounded region, we get:

$$\|(\mathcal{H}(\theta_1)^{-1} - \mathcal{H}(\theta_2)^{-1})g(\theta_2)\| \leq \frac{MG}{\mu_1\mu_2}\|\theta_1 - \theta_2\|. \tag{29}$$

Thus

$$|f(S \mid \theta_1) - f(S \mid \theta_2)| \leq \left(\frac{H}{\mu_1} + \frac{MG}{\mu_1\mu_2}\right)\|\theta_1 - \theta_2\|. \tag{30}$$

$\theta^t$ is obtained via Maximum Likelihood Estimation (MLE), we have: $\sqrt{|S_t|}(\theta^t - \theta^*) \xrightarrow{d} \mathcal{N}(0, I^{-1}(\theta^*))$, where $I(\theta^*)$ is the Fisher information matrix. This follows the asymptotic normality of MLE [44]. This implies that, with high probability, the estimation error satisfies

$$\|\theta^t - \theta^*\| \leq \frac{C(\delta)}{\sqrt{|S_t|}} \quad \text{with probability at least } 1 - \delta, \tag{31}$$

for some constant $C(\delta)$ depending on the trace and spectral norm of $I^{-1}(\theta^*)$. Since $f(S|\theta)$ is Lipschitz continuous in $\theta$, with probability at least $1 - \delta$,

$$|f(S \mid \theta^t) - f(S)| \leq \left(\frac{H}{\mu_1} + \frac{MG}{\mu_1\mu_2}\right)\|\theta^t - \theta^*\| \leq \left(\frac{H}{\mu_1} + \frac{MG}{\mu_1\mu_2}\right)\frac{C(\delta)}{\sqrt{|S_t|}} \tag{32}$$

This guarantees that for sufficiently large $|S_t|$, the error introduced by using $\theta^t$ in place of $\theta^*$ is small with high probability.

$\square$

# D    Full Derivation of Bias-Corrected Estimate

We begin by defining the perturbed objective:

$$\theta_{S(m)}^{\gamma} = \arg\min_{\theta} \frac{1}{|S|}\sum_{i \in S}\ell_i(\theta) - \gamma\ell_m(\theta) + \gamma\widetilde{\ell}_m(\theta). \tag{33}$$

Since $\theta_{S(m)}^{\gamma}$ minimizes the objective, it satisfies the first-order optimality condition:

$$0 = \frac{1}{|S|}\sum_{i \in S}\nabla\ell_i(\theta_{S(m)}^{\gamma}) - \gamma\nabla\ell_m(\theta_{S(m)}^{\gamma}) + \gamma\nabla\widetilde{\ell}_m(\theta_{S(m)}^{\gamma}). \tag{34}$$

We now apply a first-order Taylor expansion of the gradient around $\theta_S$:

$$0 \approx \frac{1}{|S|}\sum_{i \in S}\nabla\ell_i(\theta_S) - \gamma\nabla\ell_m(\theta_S) + \gamma\nabla\widetilde{\ell}_m(\theta_S)$$

$$+ \left[\frac{1}{|S|}\sum_{i \in S}\nabla^2\ell_i(\theta_S) - \gamma\nabla^2\ell_m(\theta_S) + \gamma\nabla^2\widetilde{\ell}_m(\theta_S)\right]\left(\theta_{S(m)}^{\gamma} - \theta_S\right). \tag{35}$$

Choosing $\gamma = \frac{1}{|S|}$, and noting that $\theta_S$ satisfies the original optimality condition $\sum_{i \in S}\nabla\ell_i(\theta_S) = 0$, we simplify:

$$0 \approx -\frac{1}{|S|}\nabla\ell_m(\theta_S) + \frac{1}{|S|}\nabla\widetilde{\ell}_m(\theta_S)$$

$$+ \left[\frac{1}{|S|}\sum_{i \in S\setminus q_m}\nabla^2\ell_i(\theta_S) + \frac{1}{|S|}\nabla^2\widetilde{\ell}_m(\theta_S)\right](\theta_{S(m)} - \theta_S). \tag{36}$$

Let $\mathcal{H}(S \setminus q_m, \theta_S) = \sum_{i \in S \setminus q_m} \nabla^2 \ell_i(\theta_S)$, and $\widetilde{\mathcal{H}}(q_m, \theta_S) = \nabla^2 \widetilde{\ell}_m(\theta_S)$, we obtain:

$$\theta_{S(m)} - \theta_S \approx \left[\mathcal{H}(S \setminus q_m, \theta_S) + \widetilde{\mathcal{H}}(q_m, \theta_S)\right]^{-1} \left(\nabla \ell_m(\theta_S) - \nabla \widetilde{\ell}_m(\theta_S)\right). \tag{37}$$

Now, recall the definitions of the original and flipped losses:

$$\ell_m(\theta) = -y_m \log p_\theta(q_m, 1) - (1 - y_m) \log p_\theta(q_m, 0), \tag{38}$$

$$\widetilde{\ell}_m(\theta) = -(1 - y_m) \log p_\theta(q_m, 1) - y_m \log p_\theta(q_m, 0). \tag{39}$$

Taking the gradient difference:

$$\begin{aligned}
\nabla \ell_m(\theta_S) - \nabla \widetilde{\ell}_m(\theta_S) &= \nabla \left[-y_m \log p_\theta(q_m, 1) - (1 - y_m) \log p_\theta(q_m, 0)\right] \\
&\quad - \nabla \left[-(1 - y_m) \log p_\theta(q_m, 1) - y_m \log p_\theta(q_m, 0)\right] \\
&= (1 - 2y_m) \nabla \log \frac{p_\theta(q_m, 1)}{p_\theta(q_m, 0)}. 
\end{aligned} \tag{40}$$

Substituting back, we obtain the final expression:

$$\theta_{S(m)} \approx \theta_S + \left[\mathcal{H}(S \setminus q_m, \theta_S) + \widetilde{\mathcal{H}}(q_m, \theta_S)\right]^{-1} (1 - 2y_m) \nabla \log \frac{p_\theta(q_m, 1)}{p_\theta(q_m, 0)}. \tag{41}$$

# E   Limitations and Broader Impact

Despite the promising results of CFAT, several limitations remain that open avenues for future research. For example, while CFAT incorporates analytical corrections for guessing and slipping, it assumes these behavioral perturbations follow simple, predefined patterns. In practice, examinee behavior can be more complex and context-dependent. Future work could integrate richer cognitive models or leverage response time, clickstream, or eye-tracking data to better capture behavioral variability.

CFAT offers a scalable, interpretable, and computationally efficient solution for adaptive testing, with potential for broad societal benefits:

- Democratization of High-Quality Assessment: By reducing the number of required questions and computational overhead, CFAT can enable real-time, low-cost testing in low-resource settings, such as developing countries, where infrastructure is limited.

- Fairer and More Inclusive Testing: The bias-correction mechanism in CFAT helps mitigate the influence of irregular behaviors (e.g., guessing), potentially leading to fairer assessments across diverse populations. This is particularly important in high-stakes testing scenarios, where small inaccuracies can have significant consequences.

- Privacy: CFAT does not rely on large-scale user data or extensive training, reducing the need for data collection and storage. This not only preserves user privacy but also reduces the environmental footprint of deploying large-scale AI-driven assessment systems.

