# OpenReview forum: "A Closed-Form Solution for Fast and Reliable Adaptive Testing"
_NeurIPS.cc/2025/Conference — NeurIPS 2025 poster_

### Official Review · Reviewer_W1GP · 2025-06-29

**Clarity:** 3
**Significance:** 3
**Originality:** 3
**Rating:** 4
**Confidence:** 5

**Summary:**

This paper addresses the challenge of efficient and accurate human ability estimation in adaptive testing systems, which aims to select a small subset of questions tailored to an examinee’s ability level. However, existing methods rely on computationally intensive nested optimization, which limit scalability and robustness in large-scale, real-time testing environments. In response, the author(s) propose CFAT that quantifies the estimation error as first and second-order terms. The model bridges psychometric theory (e.g., IRT) with efficient computation, making it suitable for high-stakes assessments with massive question pools and diverse examinees.

**Questions:**

**Q1.** The paper's work is constrained by using IRT as the cognitive diagnosis model (CDM). Under IRT, the loss function is $\mu$-strongly convex, which ensures the submodular property of the model and thus enables the use of greedy algorithms to find optimal questions. However, if other CDMs are adopted, such as neural network-based CDMs or the DINA model, does the submodular property still hold? Can greedy algorithms still be applied?

**Q2.** Eq. (5) is derived by performing Taylor expansion on Eq. (4), where the authors introduce a complement set $Z$. Why not directly apply Taylor expansion to Eq. (3) instead?

**Ethical Concerns:**

["NO or VERY MINOR ethics concerns only"]

**Final Justification:**

After reading the author(s)'s rebuttal (including for other reviewers), I think the (additional) experiments are good enough, which can support the current theoretical work. But the major point that makes me indecisive is the strong assumptions (e.g., submodularity). This assumption narrows general theoretical contributions. Hence, we just give a marginal acceptance rating.

**Limitations:**

The theoretical guarantees strictly rely on IRT’s convexity. For non-convex CDMs, submodularity remains unproven, limiting theoretical universality.

**Quality:**

3

**Strengths And Weaknesses:**

**Strengths**

1. The paper is easy to understand, and the proof seems to be correct.

2. They propose the first closed-form solution for adaptive testing subset selection, eliminating nested optimization and enabling direct error minimization via gradient-Hessian alignment.

**Weaknesses**

1. The $\epsilon$-submodularity proof relies critically on IRT's $\mu$-strong convexity. For non-convex CDMs (e.g., DINA, complex neural CDMs), submodularity, and thus greedy optimality lacks theoretical guarantees.

2. The design (e.g., Hessian stability term) assumes smooth and twice-differentiable losses. This may not hold for discrete CDMs (e.g., DINA) or models with non-differentiable components.

3. The author(s) assumes fixed guessing/slipping probabilities ($\pi_g$ and $\pi_s$), which may not confirm with real-world behaviors.

---

> ### Author Rebuttal · Authors · 2025-07-28
>
> We sincerely appreciate your positive feedback on the readability, theoretical clarity, and methodological contributions of our work.
>
> > **Q1**: if other CDMs are adopted, such as neural network-based CDMs or the DINA model, does the submodular property still hold? Can greedy algorithms still be applied?
>
> **A1**: That's a great and insightful question. To answer directly: when using more complex models like neural network CDMs or the DINA model, the submodular property generally does **not** hold. However, greedy algorithms can **still** be applied effectively in practice.
>
> First, we focus on IRT in the main text because it's the most widely used measurement models for adaptive testing, especially due to its simplicity and interpretability. It's the foundation for many large-scale tests like the GRE. This is why all the background, related work, and theoretical analysis in our paper are exclusively based on the IRT setting. Under IRT, the estimation objective has nice mathematical properties (e.g., strong convexity and bounded Hessian), which allow us to derive an approximate submodular structure. This is the basis for the theoretical guarantees in our Lemma and Theorem.
>
> In contrast, models like neural networks are highly non-linear and non-convex. Their loss functions are generally not convex as you said, let alone strongly convex, so the theoretical results in our paper do not apply to these models. As a result, all the theoretical analysis in the paper is based solely on the IRT model.
>
> However, greedy algorithms can still be used as **heuristics** even without submodularity guarantees. In fact, many non-convex problems in practice have been shown to exhibit behavior similar to submodular functions, at least empirically [1][2]. If we can define some notion of "informativeness" or "uncertainty", greedy selection can still be a powerful strategy, e.g., active learning, coreset. Moreover, Table 1 demonstrates that even without theoretical guarantees, greedy selection still performs well in practice when applied to neural network methods. We'll highlight this point in the next version.
>
>
> > **Q2**: The design (e.g., Hessian stability term) assumes smooth and twice-differentiable losses. This may not hold for discrete CDMs (e.g., DINA) or models with non-differentiable components.
>
>
> **A2**: Yes, the majority of the design in our paper is grounded in IRT. This is because IRT remains the dominant framework in most large-scale testing, as discussed in motivation and related work section. As you noted, for discrete CDMs, there is currently a lack of reliable theoretical foundations in this domain. While our analysis centers on IRT, we do consider complex approaches as well. For example, to calculate Hessian in complex models like NeuralCDM, computing the exact Hessian is often infeasible due to the high dimensionality and nonlinearity of the parameter space. In these cases, we use a quasi-Newton approximation of $\mathcal{H}^{-1}$, as described in Appendix A.
>
> > **Q3**: The author(s) assumes fixed guessing/slipping probabilities $\pi_g$ and $\pi_s$, which may not confirm with real-world behaviors.
>
> **A3**: Thank you for your comment. Yes. In real-world settings, guessing and slipping behaviors can vary across different examinees and items. In our work, we use fixed values for  $\pi_g$ and $\pi_s$ as a simplifying assumption to make the bias correction process more transparent and easier to interpret. Also, these probabilities are only used during the question selection stage, and not treated as item parameters that can be directly estimated. Our current approach aims to capture *general behavioral trends*, for example, $\pi_s$ reflects an overall tendency for careless mistakes across the population. We agree that adapting or learning these probabilities dynamically could improve the model's realism. In future work, we plan to explore strategies that allow us to optimize $\pi_g$ and $\pi_s$ jointly during the question selection process.
>
> > **Q4**: Eq. (5) is derived by performing Taylor expansion on Eq. (4), where the authors introduce a complement set $Z$. Why not directly apply Taylor expansion to Eq. (3) instead?
>
> **A4**: Yes, performing Taylor expansion directly on Eq. (3) is mathematically equivalent to the approach we use in Section 3.1 with the introduction of complement set $Z$. Our intention here was to convey: rather than physically removing the influence of $Z$ from the item pool, we apply a soft adjustment by down-weighting its contribution. This idea aligns with the strategy discussed later in Section 3.3, where we handle noisy samples in set $S$ by down-weighting them instead of removing/replacing them. By keeping this treatment consistent across sections, we aim to improve the overall conceptual coherence of the method. To avoid confusion, we will clarify the equivalence of these two more explicitly in the revised version.
>
> Thank you again for your careful review. If anything remains unclear, we'd be happy to discuss it further.
>
>
> Reference:
>
> [1] Krause, Andreas, and Daniel Golovin. "Submodular function maximization." Tractability 3.71-104 (2014): 3.
>
> [2] Sener, Ozan, and Silvio Savarese. "Active Learning for Convolutional Neural Networks: A Core-Set Approach." International Conference on Learning Representations. 2018.

---

> ### Author Response · Authors · 2025-08-04
> **Comment**
>
> Thanks for your positive feedback. If anything unclear, please feel free to contact us.

---

> ### Comment · Area_Chair_XmKF · 2025-08-06
> **Response to rebuttal.**
>
> Hi reviewer W1GP!
>
> Do you find the rebuttal convincing?

---

> > ### Comment · Reviewer_W1GP · 2025-08-07
> >
> > Yes, I appreciate the authors' rebuttal. As the authors' responses, I think all the theoretical works are based on the IRT model. For other CDMs, according to the authors' acknowledgment, some critical prerequisites, such as submodularity, could not hold. While IRT is widely used, this contribution may limit the generalization in adaptive learning. However, I welcome this idea and remain my rating, which may inspire other future work in adaptive learning.

---

> > > ### Author Response · Authors · 2025-08-07
> > > **Response**
> > >
> > > Thank you so much for your encouraging feedback. We agree that extending the theoretical results to other CDMs is a promising direction for future work. We're very grateful for your positive assessment and support, and it truly means a lot to us.

---

### Official Review · Reviewer_pzeL · 2025-07-03

**Clarity:** 3
**Significance:** 2
**Originality:** 2
**Rating:** 3
**Confidence:** 3

**Summary:**

CFAT efficiently selects question subsets using a closed-form objective, outperforming meta-learning and reinforcement learning methods by an average of 2% in prediction accuracy while reducing the number of required questions by at least 15%. It achieves up to 12× faster selection than BECAT and consistently chooses high-discrimination questions aligned with examinees’ abilities, offering strong interpretability and generalizability.

**Questions:**

1. Same as in the Weaknesses Section. Is there a better baseline or a dataset that we can experiment with? For example, we can think of JUNYI as a dataset.

2. The experimental results presented in the BECAT paper and the experimental results presented in this study are almost the same. However, in UAT and CCAT published at a similar time, there is a considerable difference in the baseline performance even though the same dataset and metrics are used. Why is this so? If there can be differences in each experiment, it seems that sufficient experimental data is needed (such as mean or std).

**Ethical Concerns:**

["NO or VERY MINOR ethics concerns only"]

**Final Justification:**

Quality - I find this paper well written and am satisfied with the overall quality of the paper.

Clarity - The problem that the paper is trying to solve, its motivation, and its approach are all clearly understood.

Significance - I've looked into some related studies and provided additional baselines and datasets, but overall, I feel like the task is stagnant and saturated.

Originality - The authors improve efficiency compared to existing methods by correcting for bias caused by guessing and slipping and representing estimation errors as closed-form expressions. The proposed method is intuitive and novel. However, despite the performance improvement, it does not present a significant new direction, as it adheres to the standard evaluation methods of Adaptive Testing.

Overall, this was extremely difficult to confirm, and this study has both strengths and weaknesses. The advantage is that it's a well-written, high-quality paper, but the disadvantage is that it maintains a baseline, dataset, and evaluation metrics that are largely similar to those of related studies from several years ago, making it somewhat lacking in freshness. There are reasons to accept and reasons to reject this research.

Confidence - At first, this was an unfamiliar task for me, but as I continued to research related research and communicate with authors, I became quite confident.

While the authors could have done a little more to advance the relevant tasks, the overall quality of the paper was satisfactory. Thank you.

**Limitations:**

Yes

**Paper Formatting Concerns:**

No.

**Quality:**

4

**Strengths And Weaknesses:**

Strengths

1.The purpose of the research is clear and definite. The quality of the paper is also high.

2.Not only are the experimental results excellent, but the theoretical support is also solid.

Weaknesses
1. Even considering the existing trends of the task, I have some doubts about whether it is a novel method that shows overwhelmingly good performance. I don't feel that it is a significant improvement over the existing baseline.

2. It is questionable whether any comparisons with recent studies have been made. For example, we could easily find a comparable baseline, 'Liu, Zirui, et al. "Computerized adaptive testing via collaborative ranking." Advances in Neural Information Processing Systems 37 (2024): 95488-95514.'

---

> ### Author Rebuttal · Authors · 2025-07-26
>
> Thank you very much for your high evaluation of the quality of our paper. Below are our detailed responses to your concerns regarding the experiments:
>
> >**Q1**: Even considering the existing trends of the task, I have some doubts about whether it is a novel method that shows overwhelmingly good performance. I don't feel that it is a significant improvement over the existing baseline.
>
> **A1**: This is a great point. In fact, almost **no** existing selection algorithm achieves overwhelmingly good performance in terms of ACC/AUC in adaptive testing. This is largely due to the nature of the task: these metrics are evaluated under small sample sizes (e.g., 5 or 10 items), but the theoretical upper bound (i.e., using all items) is limited by measurement model itself (e.g., IRT). For example, due to noise in student responses and the fitting capacity of IRT models, the AUC upper bound on the NIPS-EDU dataset is only around 0.79. Our method already achieves 0.78 with just 20 items, while the best of the other methods reaches only 0.76-0.77. So, further improvement is inherently constrained.
>
> As you rightly pointed out, this suggests that ACC/AUC may be **approaching saturation** for this task. For example, UATS[1] only improved upon the previous best method by 0.5%–1%, and CCAT[2] did not outperform previous SOTA methods in terms of AUC/ACC at all. Because of this, the community often turns to another metric: ability estimation error (MSE) [3], which more directly aligns with the core goal of adaptive testing: minimizing estimation error. As shown in Figure 2 of our paper, our method **achieves a 15% reduction** in the number of items needed to reach the same estimation error, which we believe is a substantial and meaningful improvement.
>
> > **Q2**: It is questionable whether any comparisons with recent studies have been made. For example, we could easily find a comparable baseline, 'Liu, Zirui, et al. "Computerized adaptive testing via collaborative ranking." Advances in Neural Information Processing Systems 37 (2024): 95488-95514.'
>
> **A2**: Yes. This is a recent and relevant work. Although we cited it in our paper, we didn't include it in the experimental comparison. The main reason is that its goal is quite different: it focuses on ranking students, while our work targets accurate ability estimation. Since their method is not optimized for estimation accuracy, we initially felt the comparison might not be entirely fair. That said, we agree it's important to be thorough. So, we've now included a comparison based on MSE across three datasets (ASSIST, NIPS-EDU, and EXAM), shown below:
>
>
>
> [Table A – Dataset ASSIST]
>
>
> | Method   | Step 5       | Step 10      | Step 15      | Step 20      |
> |----------|--------------|--------------|--------------|--------------|
> | Random   | 1.182±0.206  | 0.914±0.140  | 0.866±0.117  | 0.377±0.060  |
> | Fisher   | 0.931±0.133  | 0.691±0.082  | 0.498±0.078  | 0.360±0.038  |
> | UATS     | 0.918±0.073  | 0.780±0.097  | 0.328±0.065  | 0.208±0.033  |
> | NCAT     | 1.126±0.115  | 0.753±0.095  | 0.550±0.133  | 0.196±0.030  |
> | CCAT     | 1.027±0.187  | 0.702±0.052   | 0.492±0.082   | 0.192±0.021   |
> | **CFAT** | **0.894±0.044** | **0.473±0.057** | **0.216±0.022** | **0.074±0.037** |
>
>
> [Table B – Dataset NIPS-EDU]
>
> | Method   | Step 5       | Step 10      | Step 15      | Step 20      |
> |----------|--------------|--------------|--------------|--------------|
> | Random   | 1.283±0.200  | 0.994±0.156  | 0.583±0.094  | 0.402±0.162  |
> | Fisher   | 1.142±0.103  | 0.932±0.068  | 0.495±0.093  | 0.360±0.105  |
> | UATS     | 1.200±0.074  | 0.875±0.072  | 0.475±0.062  | 0.358±0.085  |
> | NCAT     | 1.183±0.101  | 0.872±0.064  | 0.417±0.084  | 0.264±0.106  |
> | CCAT     | 1.103±0.098  | 0.927±0.061   | 0.527±0.027   | 0.374±0.078   |
> | **CFAT** | **0.913±0.066** | **0.861±0.076** | **0.250±0.050** | **0.131±0.038** |
>
> [Table C – Dataset EXAM]
>
>
> | Method   | Step 5       | Step 10      | Step 15      | Step 20      |
> |----------|--------------|--------------|--------------|--------------|
> | Random   | 1.387±0.188  | 1.242±0.162  | 0.834±0.166  | 0.586±0.108  |
> | Fisher   | 1.182±0.176  | 0.916±0.152  | 0.681±0.072  | 0.314±0.029  |
> | UATS     | 1.178±0.121  | 0.820±0.102  | 0.449±0.093  | 0.291±0.049  |
> | NCAT     | 1.176±0.128  | 0.803±0.110  | 0.547±0.105  | 0.241±0.083  |
> | CCAT     | 1.183±0.152  | 0.888±0.096   | 0.568±0.077  | 0.283±0.024   |
> | **CFAT** | **1.018±0.058** | **0.735±0.075** | **0.332±0.051** | **0.127±0.020** |
>
>
> ---
>
> As shown, our method consistently outperforms CCAT in terms of MSE across all steps and datasets. We'll include these results in the next version of the paper for completeness. If you have any other baselines you'd recommend for comparison, we’d be more than happy to include them as well. Thanks again for your valuable feedback!
>
>
> > **Q3**:  Is there a better baseline or a dataset that we can experiment with? For example, we can think of JUNYI as a dataset.
>
> **A3**: Thank you for your suggestion. We agree that exploring additional datasets can help further validate the effectiveness and generalizability of our approach. In response, we have conducted experiments on the JUNYI dataset, and the results are presented below. We plan to include these results in the next version of the paper.
> | **Method**         | **Step@5**        | **Step@10**       | **Step@20**       |
> |--------------------|-------------------|-------------------|-------------------|
> | Random+IRT         | 71.61 / 73.94     | 73.11 / 76.54     | 75.46 / 79.90     |
> | Fisher+IRT         | 72.84 / 75.83     | 74.80 / 78.48     | 76.11 / 81.05     |
> | KL+IRT             | 72.15 / 74.59     | 74.19 / 78.42     | 75.50 / 80.98     |
> | MAAT+IRT           | 71.83 / 74.61     | 74.35 / 77.54     | 74.09 / 79.66     |
> | BECAT+IRT          | 74.57 / 79.34     | 76.02 / 80.61     | 76.89 / 81.48     |
> | NCAT+IRT           | 72.26 / 77.64     | 75.62 / 80.96     | 76.98 / 81.53     |
> | BOBCAT+IRT         | 72.22 / 78.23     | 75.48 / 80.22     | 76.98 / 81.67     |
> | **CFAT+IRT**       | **75.77 / 79.58** | **77.45 / 81.20** | **78.81 / 83.05** |
>
> ---
>
> | **Method**             | **Step@5**        | **Step@10**       | **Step@20**       |
> |------------------------|-------------------|-------------------|-------------------|
> | MAAT+NeuralCD          | 72.81 / 75.31     | 75.06 / 77.61     | 75.62 / 80.96     |
> | BECAT+NeuralCD         | 77.11 / 80.28     | 78.33 / 83.00     | 79.02 / 84.00     |
> | NCAT+NeuralCD          | 75.33 / 79.55     | 77.59 / 81.36     | 78.44 / 84.16     |
> | BOBCAT+NeuralCD        | 76.09 / 80.24     | 77.42 / 82.79     | 78.47 / 84.06     |
> | **CFAT+NeuralCD**      | **78.23 / 82.41** | **80.51 / 83.27** | **81.91 / 86.54** |
>
> > **Q4**: However, in UAT and CCAT published at a similar time, there is a considerable difference in the baseline performance even though the same dataset and metrics are used. Why is this so?
>
> **A4**: Thank you for raising this question. Yes, although UAT and CCAT were published around the same time and used the same dataset and evaluation metrics on the surface, there are notable differences in their baseline performance. This discrepancy can be attributed to several key factors:
>
> - CCAT primarily focuses on ranking the examinees, and thus adopts Ranking Consistency metrics. In addition, the optimization strategies used in CCAT differ significantly: it employs MCMC-based methods and specific gradient descent, which are tailored to their specific modeling objectives.
>
> - Data Splitting Strategies: The data partitioning schemes differ between the two works. In our study, we follow a 7:2:1 split for training, validation, and testing, while UAT adopts a 6:2:2 split. Such differences can impact the model’s generalization and reported performance.
>
> - Data Preprocessing and Filtering: UAT is a data-driven method and applies its own data filtering criteria. Based on the released code and dataset details, it appears that UAT filters out a considerable number of examinees or items with insufficient interaction records. This can lead to a cleaner but less diverse dataset, which may inflate performance metrics.
>
>
>
>
> > **Q5**: If there can be differences in each experiment, it seems that sufficient experimental data is needed (such as mean or std).
>
>
>
> **A5**: Yes, we completely agree. In our paper, Figure 2 already presents the mean and standard deviation of MSE, with results averaged over 10 repetitions and error bars indicating the standard deviation. To further enhance the transparency of our findings, we will include the std of AUC/ACC metrics in the appendix in the next version.
>
>
> Please feel free to reach out if you have any further concerns or would like to discuss in more detail!
>
> [1] Yu, Junhao, et al. "A unified adaptive testing system enabled by hierarchical structure search." Forty-first International Conference on Machine Learning. 2024.
>
> [2] Liu, Zirui, et al. "Computerized adaptive testing via collaborative ranking." Advances in Neural Information Processing Systems 37 (2024): 95488-95514.
>
> [3] Cheng, Ying. Computerized adaptive testing—new developments and applications. University of Illinois at Urbana-Champaign, 2008.

---

> > ### Comment · Reviewer_pzeL · 2025-08-05
> >
> > Thank you so much for your kind answers to my questions. I fully understand Q2, Q3, Q4, and Q5, and I am satisfied with the authors' rebuttal. In particular, the addition of a new baseline method and benchmarks seems highly desirable.
> >
> > Also, thank you for your understanding of my comments regarding Q1. I believe the quality of the paper is excellent and the proposed method is novel. My additional question is, as you answered in A1, it seems that the ACC/AUC of the task you are studying has already reached saturation. Is there any way to improve this?
> >
> > I fully understand that this study isn't about improving the evaluation method. However, while researching related studies, I noticed that because they were conducted using the same evaluation method, there wasn't a significant perceived performance improvement. While this may not be the main research point of the paper, I still wonder if there isn't a better evaluation method to demonstrate the method's superiority.
> >
> > Thank you very much.

---

> ### Author Response · Authors · 2025-08-05
> **Response**
>
> Thank you very much for the thoughtful follow-up and your encouraging words. You're right: when ACC/AUC approaches saturation, especially under short test lengths, it becomes increasingly difficult to showcase meaningful improvements. We truly appreciate you highlighting it. To address this, one approach we adopt in our paper is using Mean Squared Error (MSE) for ability estimation. As mentioned both in A1 and the main text, MSE directly reflects how accurately we estimate a student’s latent ability, which is the fundamental goal of adaptive testing. Compared to ACC/AUC, it tends to be more sensitive to improvements in selection strategies.
>
> Beyond MSE, we believe there are several promising directions worth exploring:
>
> - Designing more realistic evaluation settings, such as testing robustness and stability under perturbations. For example, in our paper, we experiment with noise injection (Table 5) to evaluate how well different methods hold up under uncertainty. This helps reveal the more fundamental strengths of each approach.
>
> - Improving the capacity of the measurement model itself. As noted in A1, the theoretical performance ceiling is often limited by the expressiveness of the model (e.g., IRT). Developing more powerful or flexible measurement models could raise the upper bound of achievable AUC, even when using all available data. But, measurement modeling is itself a distinct research area, and advancing it requires broader community collaboration.
>
>
> - Involving human experts in the evaluation loop. Though expensive, deploying systems in real or simulated environments with expert judgment can provide valuable insights into how meaningful and appropriate the selected questions are. Some recent works in personalized learning have taken steps in this direction.
>
> These ideas are still under exploration and haven’t yet become standardized in the field, but we are committed to contributing toward building better evaluation benchmarks. We look forward to hearing more thoughts from you and please don’t hesitate to continue the discussion if you have further suggestions.

---

> ### Comment · Reviewer_pzeL · 2025-08-07
>
> Thank you for your response. Let me be a little more specific about my question.
>
> For example, from what I know about Active Learning (I think MAAT is one of the baselines), there is a certain trade-off between uncertainty and diversity. Accordingly, in Active Learning research, uncertainty-based methods record good performance, but they have the limitation that they may miss the diversity of the overall distribution. Of course Acc is also measured, but a diversity-based metric is presented to compare performance from various angles.
>
> So my question is: Is there a way in Adaptive Testing to measure the similarity to the overall distribution or the diversity of the selected data? From what I've found, previous studies have presented performance in a similar way. Are there any reasons or limitations that make it so standardized?

---

> ### Author Response · Authors · 2025-08-07
> **Response**
>
> Thank you so much for the follow-up. This is a really thoughtful point. Yes, in Active Learning, there's often a trade-off between uncertainty and diversity. However, we believe diversity hasn’t become a standard metric in Adaptive Testing because the goal here is different. In most cases, adaptive testing is focused on estimating a student’s latent ability as accurately and efficiently as possible. For that, information-based methods (like Fisher or KL) are more directly tied to reducing estimation error. These approaches are backed by well-established theory from psychometrics, where minimizing uncertainty in the estimate is the main objective [1].
>
> Diversity, on the other hand, doesn’t always help with that. Here’s a simple example to illustrate:
>
> > Imagine you have a question pool with questions ranging from very easy to very hard, say difficulty levels from 0.0 to 1.0. Now, suppose you pick questions that are evenly spaced out to “cover” the whole range (that would look diverse). But let’s say one student’s ability is around 0.7. In that case, asking questions that are way too easy or way too hard isn’t very informative (waste of time). The most helpful questions are actually those close to their ability level, maybe between 0.6 and 0.8. So selecting based on uncertainty/informativeness, i.e., choosing questions where the model is most unsure actually leads to better estimation in practice. You can think of it a bit like *binary search* algorithm: we get the most value by focusing around the target iteratively.
>
> So while diversity is definitely an interesting angle, it *doesn’t align with the core measurement goal of adaptive testing* (i.e., accurate ability parameter estimation). That’s probably why it hasn’t been widely used as a standard evaluation metric in this field.  In contrast, metrics like MSE directly reflect the error in ability estimation, while AUC/ACC evaluate how well we can predict student responses (which indirectly indicate how accurate our ability estimates are) [1].
>
>
>
> We really appreciate you raising this point and it's a valuable perspective. If you have more ideas, we’d love to hear them and keep the conversation going.
>
> Reference:
>
> [1] Liu, Qi, et al. "Survey of computerized adaptive testing: A machine learning perspective." arXiv preprint arXiv:2404.00712 (2024).

---

### Official Review · Reviewer_RhGM · 2025-07-05

**Clarity:** 3
**Significance:** 3
**Originality:** 3
**Rating:** 5
**Confidence:** 2

**Summary:**

This paper addresses the approximation limitations of existing approaches to adaptive testing, which often require iterative computations and hinder their applicability in online settings. To overcome this, the authors propose a closed-form solution for question selection that supports efficient ability estimation with bias correction. The method is evaluated across four dimensions: (1) student performance prediction, (2) ability estimation accuracy, (3) computational efficiency, and (4) reliability. Experimental results on three datasets show that the proposed approach outperforms all baseline methods in most cases.

**Questions:**

* Table 1: To improve clarity, it would be helpful to explicitly label the variants of the proposed method in the table (e.g., "CFAT + IRT" and "CFAT + NeuralCDM") so that readers can easily distinguish the configurations being evaluated.

* In the evaluation section, Figure 2 presents results for the ability estimation and computational efficiency tasks but includes only a subset of the baseline methods. Could the authors clarify the rationale for selecting these specific baselines, and why the others were omitted?

**Ethical Concerns:**

["NO or VERY MINOR ethics concerns only"]

**Limitations:**

yes

**Quality:**

3

**Strengths And Weaknesses:**

Motivation: The paper clearly identifies a relevant research gap and provides a strong motivation for addressing the computational limitations of adaptive testing in online settings.

Literature Review: The related work section effectively introduces the necessary background and the context of existing methods.

Methodology: The proposed approach is well-designed and thoughtfully constructed. The theoretical analysis and proofs included in the paper provide solid support for the method's validity.

Evaluation: The proposed method is thoroughly evaluated on three datasets against eight baseline models, including random selection, classical approaches, and a variety of learning-based (e.g., active learning, meta-learning, reinforcement learning, and graph neural networks) and heuristic methods. The results show that the proposed method consistently outperforms these baselines in both student performance prediction and ability estimation tasks, whether integrated with classical IRT models or neural network-based approaches. Furthermore, the method demonstrates better computational efficiency, relatively effective alignment between selected questions and examinee ability distributions, and robust reliability under response noise caused by guessing and slipping behaviors.

---

> ### Author Rebuttal · Authors · 2025-07-25
>
> Thank you very much for your positive feedback on the motivation, methodology, and evaluation of our work. This is truly encouraging our commitment to further exploring this direction. We have carefully considered all your suggestions and have made following responses:
>
> > **Q1**: Table 1: To improve clarity, it would be helpful to explicitly label the variants of the proposed method in the table (e.g., "CFAT + IRT" and "CFAT + NeuralCDM") so that readers can easily distinguish the configurations being evaluated.
>
> **A1**: Thank you for pointing this out. To avoid confusion and improve clarity, we will revise Table 1 in the next version by merging the two sub-tables and explicitly labeling all method combinations, such as "XXX + IRT" and "XXX + NeuralCDM". This labeling will make it easier for readers to distinguish the configurations and better understand the comparative performance of each method.
>
> > **Q2**: In the evaluation section, Figure 2 presents results for the ability estimation and computational efficiency tasks but includes only a subset of the baseline methods. Could the authors clarify the rationale for selecting these specific baselines, and why the others were omitted?
>
> **A2**: In the evaluation section, we chose to include only a representative subset of baseline methods in Figure 2 due to space limitations. Specifically, we selected one widely recognized method from each category, following the taxonomy in [1]: Fisher for traditional information-based methods, MAAT for active learning, NCAT for data-driven approaches, BECAT for subset selection, and CFAT as our proposed method. Within each category, the runtime differences among methods are generally minor. For completeness, we've collected the full runtime statistics across all baselines as follows:
>
> | Method   | Time (s) |
> |----------|--------------------|
> | Fisher   | 0.23177            |
> | Random   | 0.25821            |
> | KL       | 0.79015            |
> | BECAT    | 4.38283            |
> | MAAT     | 148.23482          |
> | NCAT     | 19.58382            |
> | BOBCAT   | 0.69179            |
> | UATS     | 33.74935           |
> | **CFAT**     | 0.35180            |
>
>
> As shown, CFAT achieves a strong balance between efficiency and accuracy. It runs only slightly slower than Fisher, yet delivers SOTA performance in ability estimation. Based on your suggestion, we will include the complete runtime statistics in the appendix of the next version.
>
>
>
>
> We truly appreciate your suggestion. Please feel free to share any further insights or suggestions you might have.
>
> Reference:
>
> [1] Liu, Qi, et al. "Survey of computerized adaptive testing: A machine learning perspective." arXiv preprint arXiv:2404.00712 (2024).

---

> > ### Comment · Area_Chair_XmKF · 2025-08-06
> > **Response to rebuttal.**
> >
> > Hi reviewer RhGM,
> >
> > Do you find the author response convincing?

---

> > ### Comment · Reviewer_RhGM · 2025-08-07
> >
> > Thank you for your detailed and thoughtful responses. I appreciate the clarifications and satisfied with the responses and have no further questions.

---

> > > ### Author Response · Authors · 2025-08-08
> > > **Response**
> > >
> > > Thank you so much for your positive evaluation and are glad the responses were helpful. Your support and time mean a lot to us!

---

### Official Review · Reviewer_Hxwf · 2025-07-08

**Clarity:** 3
**Significance:** 2
**Originality:** 2
**Rating:** 4
**Confidence:** 3

**Summary:**

The authors proposed a closed form solution to tackle the question selection problem in Adaptive Testing systems. The detailed theoretical analysis is provided and experiments demonstrate the effectiveness of the proposed method.

**Questions:**

See above weaknesses.

**Ethical Concerns:**

["NO or VERY MINOR ethics concerns only"]

**Final Justification:**

Based on the discussions, I would vote for a borderline accept for this work. The proposed method is neat, with solid theoretical analysis, however, the effectiveness of this method in real world application is still under investigation.

**Limitations:**

See above weaknesses.

**Quality:**

2

**Strengths And Weaknesses:**

Strengths:
1. This work is well motivated.
2. Related work are well studied and discussed.
3. The idea is neat, with extensive theoretical support.
4. The experimental results demonstrate the improvement of the proposed method.

Weaknesses:
1. The reviewer's first question lies in the invertible assumption of Hessian matrix. Since the problem setup is to find a subset of questions that can reflect students' ability across all questions, it indicates a low rank assumption. In such cases, how could we assume the Hessian matrix is invertible, and thus leads to the closed form solution?
2. The datasets used in the experiment section are not well introduced, e.g., statistics, distribution.
3. It's not clear how the method can be used in large scale applications.

---

> ### Author Rebuttal · Authors · 2025-07-25
>
> Thank you for your positive feedback on the motivation, core idea, and experimental results of our paper. Here are the responses to each of your questions:
> > **Q1**: ...How could we assume the Hessian matrix is invertible?
>
> **A1**: We appreciate the opportunity to clarify. In the context of adaptive testing, **this assumption is actually standard and reasonable**. As mentioned in Section 3.1, line138 of our paper:  *"$\mathcal{H}(S, \theta)$ is invertible, which holds under standard regularity conditions in IRT. For complex neural network-based models, where computing the exact Hessian is often intractable, we adopt a quasi-Newton approximation (details are provided in Appendix A)."*
>
> Let us explain this in more detail:
>
> - The invertibility of the Hessian is a widely accepted assumption in adaptive testing, often referred to in psychometric literature as "identifiability and estimability under local independence and monotonicity assumptions" [1]. In fact, under standard regularity conditions of IRT models, the loss function $\ell(\theta)$ is strongly convex [2][3], which **ensures that the Hessian $\mathcal{H}$ is positive definite and thus invertible**. So in classical IRT settings, this assumption is well-supported both theoretically and empirically. More importantly, all large-scale standardized tests today (e.g., GRE) are fundamentally based on IRT, which further motivates the paper’s focus on IRT-based frameworks.
>
> - For more complex models like neural network approaches (e.g., NeuralCDM), computing the exact Hessian is often intractable due to the high dimensionality and nonlinearity of the parameter space. Therefore, in these cases, we adopt **a quasi-Newton approximation of $\mathcal{H}^{-1}$, as detailed in Appendix A**. This allows us to iteratively approximate the inverse Hessian without computing it directly, which makes the approach computationally feasible while still preserving the theoretical foundation of using $\mathcal{H}^{-1}$ for subsequent analysis.
>
> To avoid confusion, we'll make this assumption and its justification more explicit in the revised version. We hope this could address your concern.
>
>
>
> > **Q2**: The datasets used in the experiment section are not well introduced, e.g., statistics, distribution.
>
> **A2**: Thank you for pointing this out. Below is a summary of the key statistics for the datasets used in our experiments:
>
> | Dataset       | ASSIST     | NIPS-EDU   | EXAM       |
> |---------------|------------|------------|------------|
> | #Students     | 20,704     | 123,889    | 96,329      |
> | #Questions    | 15,071     | 28,561     | 1,714          |
> | #Responses    | 1,768,253  | 17,250,577  | 892,676    |
> | Responses/Student | 85.41  | 139.24      | 9.27      |
> | Responses/Question | 117.33 | 603.99    | 520.81      |
>
>
> We will also provide additional visual analyses of the datasets (e.g., the distribution of the number of interactions per student), which are difficult to present here. We apologize for the oversight and all detailed dataset information and visualizations will be included in the appendix in the revised version.
>
> > **Q3**: It's not clear how the method can be used in large scale applications.
>
> **A3**: That’s a great question. Our method is actually well-suited for large-scale applications for two main reasons:
>
> 1. No training required: Our approach is based on a simple greedy algorithm and does not involve any model training. This makes it significantly more efficient than training-heavy methods like reinforcement learning or active learning. As shown in Figure 2(b), our method achieves at least 12× speedup in runtime.
>
> 2. Validated on large-scale datasets: We've tested our method on very large datasets  (Table 1), such as NIPS-EDU, which includes over 220,000 students and 19 million response logs. Moreover, our method remains robust even under high-noise conditions in realistic testing environments (Table 2).
>
> We appreciate your suggestion and we will highlight these advantages more clearly in the conclusion section in the next version, especially in the context of large-scale deployment. If anything's unclear, we'd be happy to discuss them.
>
>
> Reference:
>
> [1] Lord, Frederic M. Applications of item response theory to practical testing problems. Routledge, 2012.
>
> [2] Zhuang, Yan, et al. "A bounded ability estimation for computerized adaptive testing." Advances in Neural Information Processing Systems 36 (2023): 2381-2402.
>
> [3] Yu, Junhao, et al. "A unified adaptive testing system enabled by hierarchical structure search." Forty-first International Conference on Machine Learning. 2024.

---

> > ### Comment · Reviewer_Hxwf · 2025-08-05
> >
> > Thanks for the detailed response. The reviewer's 1st question lies more on the conflict part between your low rank assumption (selecting fewer, targeted questions) and the explicit usage of the invertibility of the Hessian. One more question in experiments, from the data statistics you provided, Responses/Student on EXAM dataset is only 9.27, thus, are the results reported in the EXAM columns @10, @20 from Table 1 still reliable?

---

> ### Author Response · Authors · 2025-08-06
> **Response**
>
> Thank you very much for the insightful follow-up. I fully understand your concern — there does appear to be a potential tension between the low-rank nature of the task (i.e., selecting a small number of informative questions) and the assumption of Hessian invertibility. We’d like to clarify that while our goal is to select a compact subset, this does not imply that the Hessian of loss wrt $\theta$ is low-rank: For simpler models like IRT, the invertibility of the Hessian is well-established under standard regularity conditions such as local independence and monotonicity. This theoretical foundation supports its use in many information-based strategies. For more complex models (especially neural architectures), invertibility is harder to guarantee, as you rightly pointed out. That’s why, as mentioned in A1, we adopt quasi-Newton approximations of the inverse Hessian in our paper, which provide a computationally feasible alternative while preserving the core principle.
>
> Moreover, even when the Hessian becomes ill-conditioned or non-invertible (e.g., when very few items are selected using complex NN model), our method still acts as a heuristic information-based strategy. It can be viewed as a greedy algorithm that iteratively selects items contributing the most to the information function, even if the theoretical foundation of invertibility is no longer strictly valid at early stages. The experimental results in this paper can prove this. We’ll make sure to clarify this assumption more explicitly in the revised manuscript. Your comment has been very helpful.
>
> As for your second question about the EXAM dataset (Avg. 9.27 responses/student) and the reported results at @10 and @20:
>
> - While the average number of responses is 9.27, this figure includes a long tail of students with very sparse responses. In our experiments, we filter out students who have fewer than the required number of observed responses when evaluating @10 and @20 metrics (like previous works). This ensures that the reported results are based on sufficient data and are meaningful for comparison across methods.
>
> - Additionally, the EXAM dataset is commonly used in prior adaptive testing studies precisely because it reflects a real-world low-resource scenario: many students only complete one test session, typically containing 10–20 questions. Evaluating performance on such sparse data is still important, particularly for metrics like @5, which remains reliable even under limited observations. We’ll make sure to clarify this preprocessing step in the revision to avoid any potential confusion.
>
> Thank you again for your thoughtful and constructive questions, and they’ve helped improve the clarity and rigor of our work. Please feel free to continue the discussion if anything remains unclear.

---

> > ### Comment · Area_Chair_XmKF · 2025-08-06
> > **Follow up on response.**
> >
> > Hi reviewer Hxwf,
> >
> > Do you find the authors' remarks on Hessian invertibility convincing?

---

### Comment · Area_Chair_XmKF · 2025-08-06
**Comparison with Bayesian approaches.**

Hi authors!

To me, one question I have is whether it makes sense to compare to a Bayesian baseline (where we have a Bayesian belief / probability distribution reflecting our state of knowledge about $\theta$ in equation (2) and you maximise some sort of information gain).

While I don't know much about your application (adaptive testing), this would appear to be a natural solution to the problem.

Authors: do fee free to argue this approach is misguided if that's how you feel about it. The lack of emphasis on Bayesian modelling strikes me as important given the focus of the paper is sample efficiency.

Reviewers: would you like to see a the method contrasted with a Bayesian approach?

Thanks,

area chair

---

> ### Author Response · Authors · 2025-08-07
> **Response to AC**
>
> Thanks for your thoughtful comment, and we appreciate how deeply you've engaged with the paper. YES. Bayesian methods are a very natural fit for adaptive testing, and there’s actually a fair amount of traditional psychometric work using Bayesian ideas [1]. However, in many existing papers [2–5], these approaches aren’t typically included as baselines. There are a couple of practical reasons for that:
>
> 1. Most Bayesian methods are still built on top of traditional information-based strategies (like Fisher or KL baseline in our paper). They just incorporate uncertainty modeling over the ability parameter $\theta$. Because of this, prior works often treat Fisher and KL as representative of the broader class of information-based methods.
>
> 2. More importantly, Bayesian methods can be much more computationally expensive. Many of them require integrating over the posterior (sometimes even multiple times per question), which makes them less suitable for real-time applications.
>
> But we really appreciate your suggestion, and we agree that it will be helpful to include those variants. Thus, we ran additional experiments on the ASSIST dataset with two classic Bayesian selection strategies:
>
> - Bayesian Fisher: After each response, we update the posterior over ability as $
>   p(\theta \mid \text{responses}) \propto p(\theta) \prod_{i \in S} p(y_i \mid q_i, \theta)$, and then select the question with the highest expected Fisher information $I$:
>   $$
>   q^* = \arg\max_q \int I(q, \theta)\, p(\theta \mid \text{responses}) d\theta
>   $$
>
> - Bayesian KL: This method chooses the question that leads to the greatest expected change in the posterior distribution:
> $$
>   q^* = \arg\max_q E_{y} [D_{KL}(p(\theta | responses, y_q) || p(\theta |{responses})) ]
>   $$
>
> Below are the MSE results across different test lengths:
>
> | Method       | Step 5        | Step 10       | Step 15       | Step 20       |
> | ------------ | ------------- | ------------- | ------------- | ------------- |
> | Random       | 1.182 ± 0.206 | 0.914 ± 0.140 | 0.866 ± 0.117 | 0.377 ± 0.060 |
> | Fisher       | 0.931 ± 0.133 | 0.691 ± 0.082 | 0.498 ± 0.078 | 0.360 ± 0.038 |
> | Fisher-Bayes | 0.923 ± 0.128 | 0.675 ± 0.077 | 0.472 ± 0.070 | 0.343 ± 0.036 |
> | KL           | 0.972 ± 0.145 | 0.658 ± 0.085 | 0.475 ± 0.072 | 0.362 ± 0.040 |
> | KL-Bayes     | 0.945 ± 0.139 | 0.629 ± 0.079 | 0.441 ± 0.068 | 0.351 ± 0.039 |
> | UATS         | 0.918 ± 0.073 | 0.780 ± 0.097 | 0.328 ± 0.065 | 0.208 ± 0.033 |
> | NCAT         | 1.126 ± 0.115 | 0.753 ± 0.095 | 0.550 ± 0.133 | 0.196 ± 0.030 |
> | CCAT         | 1.027 ± 0.187 | 0.702 ± 0.052 | 0.492 ± 0.082 | 0.192 ± 0.021 |
> | CFAT (ours)  | 0.894 ± 0.044 | 0.473 ± 0.057 | 0.216 ± 0.022 | 0.074 ± 0.037 |
>
> And here’s a rough comparison of the time cost per question:
>
> | Method       | Time per question (second) |
> | ------------ | --------------------- |
> | Fisher       | 0.23177               |
> | KL           | 0.79015               |
> | Fisher-Bayes | 5.32864               |
> | KL-Bayes     | 5.93723               |
> | CFAT	 | 0.35180  |
>
> Due to time constraints in rebuttal, we’ve only tested this on ASSIST so far. As you can see, the Bayesian variants do offer slight improvements over their non-Bayesian counterparts, especially in early test stages. As the test progresses, there’s still a gap between these and more data-driven methods like UATS, and the computational cost is higher. However, including these Bayesian baselines does make the experimental comparison more complete, and we appreciate the suggestion.
>
>
> Again, thank you for being such a responsible AC. Your suggestion is very valuable and we’ll be sure to include these comparisons in our revised version. If anything here is unclear or you have more ideas to share, we’d really love to keep the conversation going.
>
>
>
> Reference:
>
> [1] Liu, Qi, et al. "Survey of computerized adaptive testing: A machine learning perspective." arXiv preprint arXiv:2404.00712 (2024).
>
> [2] Wang, Hangyu, et al. "GMOCAT: A graph-enhanced multi-objective method for computerized adaptive testing." KDD 2023.
>
> [3] Liu, Zirui, et al. "Computerized adaptive testing via collaborative ranking." NeurIPS 2024.
>
> [4] Zhuang, Yan, et al. "A bounded ability estimation for computerized adaptive testing." NeurIPS 2023.
>
> [5] Yu, Junhao, et al. "A unified adaptive testing system enabled by hierarchical structure search." ICML 2024.

---

### Note · Authors · 2025-08-11

Hello AC and Reviewers,

Thank you for your responsible and timely follow-ups. We sincerely thank all the reviewers for recognizing our work. It has been incredibly encouraging, especially the positive feedback on the paper’s theoretical contributions: “The idea is neat, with extensive theoretical support.” (**Hxwf**), “The theoretical analysis and proofs provide solid support for the method's validity.” (**RhGM**), “The quality of the paper is also high, the theoretical support is also solid.” (**pzeL**), and “easy to understand, and the proof seems to be correct.” (**W1GP**).

After multiple rounds of detailed interactions with the reviewers, we believe we have addressed all concerns. However, for reviewer **Hxwf**,  we waited until the last minute but still received no final confirmation about his/her concern (we completely understand that this may be due to time constraints). We just wanted to check whether there are any remaining concerns, and greatly appreciate it if AC could help follow up and confirm.



Thank you again for your time and support throughout this process.

Sincerely,

The Authors

---

### Decision · Program_Chairs · 2025-09-17

**Decision:**

Accept (poster)

**Comment:**

The paper studies an interesting problem: given a (computerised) test give to a human, how to pick questions out of a predefined set in order to elicit as much information about the human's ability as possible.

The main strength of the paper is the practicality of the proposed method. Greedy optimisation of a closed form objective is easy to implement and understand.

The main weakness is that the assumptions made in the theoretical part are not very realistic (see review W1GP).

I lean towards acceptance because (1) the task is important and has potential for a lot of positive impact in education and (2) making idealistic assumptions is quite common in the ML community.

The authors were responsive the rebuttal phase, addressing both the reviewers' concerns and my request to compare with Bayesian approaches. The discussions with reviewers were focussed on (1) experimental evaluation (datasets / baselines), (2) the suitability of assumptions.